# Decentralized trust optimization in VANETs: A blockchain-driven hybrid PoS-PBFT architecture for enhanced security and energy-efficient communication

Asad Ullah[1], Zia Ullah[2], Sanam Shahla Rizvi[3], Ibrar Ali Shah and[2], Se Jin Kwon[4]*

**1** School of Information Engineering, Xi'an Eurasia University, Xi'an, Shaanxi, China, **2** Department of Computer Software Engineering, University of Engineering and Technology Mardan, Mardan, Pakistan, **3** Raptor Interactive (Pty) Ltd., Centurion, South Africa, **4** Department of Computer Science and Artificial Intelligence, Dongguk University, Seoul, South Korea

\* skwon1109@gmail.com

## Abstract

Vehicular Ad Hoc Networks (VANETs) are essential for the success of Intelligent Transportation Systems (ITS), providing real-time communication between vehicles and infrastructure. However, the highly dynamic and decentralized nature of VANETs introduces significant challenges in ensuring trust and security across the network, including security threats, communication overhead, and energy inefficiencies. This paper presents a novel blockchain-based trust management framework that addresses these issues by incorporating lightweight consensus mechanisms, optimized data propagation strategies, and energy-aware protocols. Our approach reduces communication overhead by selectively propagating trust updates, leading to a 35% decrease in overall network traffic compared to traditional broadcast-based systems. In terms of trust accuracy, our model achieves over 95% accuracy in detecting malicious nodes, significantly outperforming existing solutions.The proposed system demonstrates the identification and penalization of malicious behaviors such as Sybil attacks and false reporting with a 25% improvement in detection rate, while maintaining low latency (an average reduction of 30% compared to PoW-based systems) and efficient energy consumption, reducing energy use by up to 40%. The proposed model also incorporates a hybrid Proof of Stake (PoS) and Practical Byzantine Fault Tolerance (PBFT) consensus mechanism, which further enhances its scalability and fault tolerance. Simulation results show that our framework converges to accurate trust values faster than traditional methods, ensuring that reliable trust evaluations are made in real-time, even under high mobility conditions. The combination of these optimizations ensures that our framework is not only secure but also highly efficient, capable of supporting scalable and resilient VANET deployments. Furthermore, our decentralized approach ensures that trust decisions are made in real-time without the need for a centralized authority, making the system more adaptable to

**Data availability statement:** All relevant data supporting the findings of this study are included within the paper.

**Funding:** This work was supported by the Basic Science Research Program through the National Research Foundation of Korea (NRF) funded by the Ministry of Education under Grant RS-2023-00244091. This work was also supported by the Technology Innovation Program (RS-2024-00507228, Development of process upgrade technology for AI self-manufacturing in the cement industry) funded By the Ministry of Trade, Industry & Energy (MOTIE, Korea). Funder: Prof. Se Jin Kwon (Department of Computer Science and Artificial Intelligence, Dongguk University, Seoul 04620, South Korea).

**Competing interests:** The authors have declared that no competing interests exist.

the high-mobility conditions of VANETs. This research offers a comprehensive solution for VANETs trust management, significantly improving communication efficiency, trust accuracy, and energy consumption while maintaining robust security and scalability. Our proposed blockchain-based trust management system provides a secure, energy-efficient, and scalable solution for VANETs, setting the stage for future developments in secure vehicular communication networks.

## 1 Introduction

Intelligent Transportation Systems (ITS) rely heavily on vehicular ad hoc networks (VANETs) to facilitate communication between vehicles and with roadside infrastructure [1,2]. It improves the safety, efficiency, and overall experience of driving through improved communication. However, the dynamic nature and decentralized nature of VANETs make ensuring the integrity of the information exchanged particularly difficult [3,4]. In VANETs, trust management is essential to mitigate risks associated with malicious behavior, misinformation, and potential cyberattacks.

As vehicles are mobile and network connections are transient, traditional security approaches are less effective in VANETs, which require robust trust management mechanisms [5]. VANET topologies are highly dynamic because vehicles often join and leave the network [6]. Due to this dynamism, trust among participating nodes is difficult to establish and maintain, as there is no permanent central authority [7]. Moreover, VANETs are vulnerable to various types of attacks, such as Sybil attacks, where a malicious node assumes multiple identities to manipulate trust evaluations [8,9]. These attacks can lead to the propagation of false information, causing accidents or traffic disruptions. To counter these threats, trust management systems must be designed to quickly and accurately assess the trustworthiness of each node based on its behavior and interactions with other nodes [10].

Fig 1 shows the current Vehicular Ad-Hoc Network (VANET) model, showing Vehicle-to-Vehicle (V2V) and Vehicle-to-Infrastructure (V2I) communication via Road Side Units (RSUs) and Internet connectivity for real-time traffic management [5]. However, this system faces significant challenges, including security vulnerabilities that leave it open to cyber-attacks, a lack of trust management that risks the spread of false information, privacy concerns due to potential exposure of sensitive data, scalability issues as vehicle numbers increase, and signal interference from environmental factors [11].

Blockchain technology has emerged as a promising solution to address the challenges of trust management in VANETs [12,13]. Blockchain technology offers higher levels of security and transparency due to its decentralized and immutable nature. In a blockchain-based VANETs, trust-related data is recorded on a distributed ledger, which is accessible to all participating nodes. This ensures that trust decisions are made transparently and that recorded trust values cannot be altered [14]. However, the integration of blockchain into VANETs is not without its challenges.

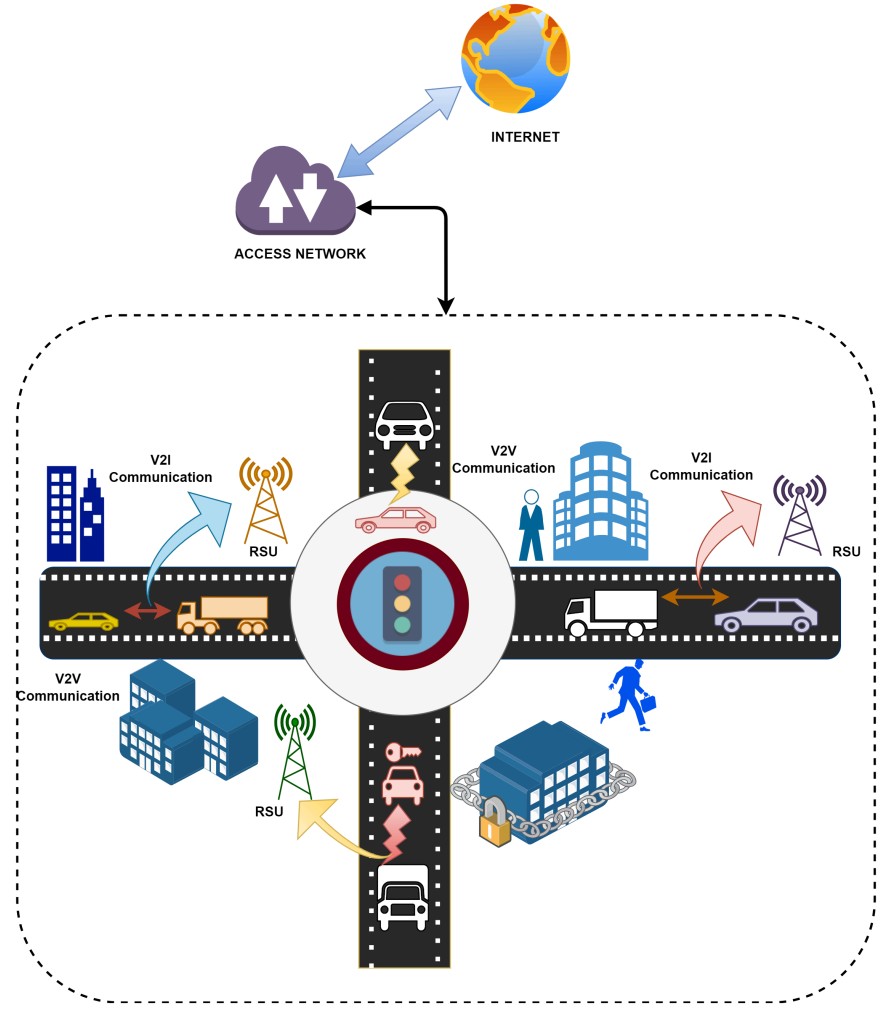

**Fig 1**. **Illustration of V2V and V2I communication through RSUs and access networks linked to the internet for VANETS.** Current issues include security vulnerabilities, lack of trust management, privacy risks, scalability challenges, and signal interference.

The integration of blockchain in VANETs introduces several challenges, particularly in terms of computational overhead, latency, and energy consumption [15,16].

Network overhead in VANETs is a critical challenge, particularly in the context of trust management, where the dynamic and highly mobile nature of the network requires frequent updates of trust evaluations, leading to significant communication and computational burdens [12]. The continuous need to exchange, verify, and record trust values between nodes increases communication overhead, which can degrade overall network performance. Moreover, in blockchain-based trust management systems [3], the consensus mechanisms required to validate and record transactions further contribute to network overhead, particularly when using computationally intensive methods such as Proof of Work (PoW) [17], which require substantial communication and processing power, leading to increased latency and energy consumption [18].

Table 1 compares PoW, PoS, PBFT, and DPoS consensus mechanisms for blockchain-based VANET trust management, focusing on computational overhead, energy consumption, and latency. PBFT excels with low values across all metrics, while PoS and DPoS offer a balance of scalability and efficiency, suitable for ITS [2,16].

**Table 1**. Comparison of consensus mechanisms in blockchain-based trust management for VANETs.

| Consensus Mechanism | Computational Overhead | Energy Consumption | Latency |
|---|---|---|---|
| PoW [16] | High | High | High |
| PoS [19] | Moderate | Low | Moderate |
| PBFT [2] | Low | Low | Low |
| DPoS [17] | Moderate | Low | Moderate |

A blockchain-based VANET also suffers from latency issues [20]. The time required to reach consensus and propagate blocks across the network can introduce delays, which are particularly problematic in time-sensitive applications such as collision avoidance and emergency response systems [2]. These latency issues can undermine the effectiveness of trust management in ensuring timely and accurate decision-making [1]. Furthermore, the energy consumption associated with maintaining a blockchain network is a significant concern [21]. Traditional blockchain networks, especially those utilizing PoW, are notorious for their high energy demands.

In a VANET environment, where energy efficiency is crucial, this level of consumption can be unsustainable and may deter widespread adoption of blockchain-based trust management systems [22]. Recent research has investigated lightweight consensus mechanisms to reduce network overhead, including Proof of Stake (PoS) and Practical Byzantine Fault Tolerance (PBFT) [9]. It is less computationally intensive than traditional Proof of Work, and it reduces the network's communications and processing load. As a result of smart contracts integrating into blockchains, trust management processes can be automated, thereby minimizing network overhead while maintaining high levels of security and trust. The VANET environment must be optimized to make blockchain-based trust management more suitable for these challenges [3,23].

This paper proposes a blockchain-based trust management framework specifically designed for VANETs to address these critical challenges. To reduce computational overhead and latency, the framework combines practical byzantine fault tolerance (PBFT) and proof-of-stake (PoS) mechanisms. By selectively propagating trust-related data only to relevant nodes, our model reduces network overhead by significantly reducing unnecessary communication. A dynamically adjusted trust management protocol for blockchain networks that is energy-aware has been developed that recognizes the energy constraints of vehicular environments. Furthermore, the innovation extends the operational lifetime of the network and ensures consistent trust accuracy. With smart contracts integrated into the framework, key trust management processes are automated, reducing the communication burden and enhancing the reliability of trust evaluation. With our contribution, VANETs will be more reliable and scalable as they deploy secure and efficient trust management systems.

## 1.1 Motivation

Vehicle Ad Hoc Networks (VANETs) are becoming more prevalent as a fundamental component of Intelligent Transportation Systems (ITS) [1,24]. When vehicles share information in real-time, it is essential to make sure it is trusted in order to prevent malicious behavior, misinformation, and cyberattacks that could compromise road safety. VANETs are highly dynamic and decentralized, so traditional centralized trust management is insufficient. As a result, trust relationships are volatile, and single points of failure are increased [17,25].

Our design is motivated by the need for decentralized, tamper-resistant, and auditable trust mechanisms in dynamic vehicular environments, requirements that traditional reputation systems often fail to meet due to their reliance on fixed infrastructure or centralized trust anchors [26,27]. Decentralized and immutable blockchain technology offers a promising solution to these problems [22,28,29]. In addition, the use of traditional consensus mechanisms such as Proof of Work (PoW), increases network overhead, computational burden, and latency. In order to leverage blockchain strengths while mitigating its weaknesses, it is imperative to explore more efficient trust management frameworks.

To address the critical issues of decreasing network overhead, reducing computational and energy costs, and reducing latency while maintaining VANET communications security and reliability, an optimized blockchain-based trust management model was developed in this study [4,10,30]. This research aims to contribute to the broader effort of enhancing blockchain technology viability in VANETs by integrating lightweight consensus mechanisms with efficient data management strategies.

## 1.2 Contributions

The primary contributions of this paper are outlined as follows:

- **Optimized Blockchain-Based Trust Management Framework:** To manage trust specifically in Vehicular Ad Hoc Networks (VANETs), the proposed framework utilizes blockchain technology. To improve network performance in high-mobility vehicular environments, the framework integrates lightweight consensus mechanisms such as Practical Byzantine Fault Tolerance (PBFT) and Proof of Stake (PoS).
- **Decentralized Trust Evaluation with Reduced Network Overhead:** A decentralized trust evaluation model is proposed, effectively distributing trust management processes across multiple nodes. Decentralization reduces the risk of single points of failure by eliminating dependency on a central authority. Furthermore, the model introduces a mechanism for selectively propagating trust-related data by only distributing it to nodes directly involved in the transaction or interaction.
- **Energy-Efficient Trust Management:** The proposed framework incorporates an energy-aware trust management protocol to address energy consumption challenges in resource-constrained vehicular environments. Using this protocol, participants in the blockchain network are dynamically adjusted according to their energy resources, thus extending the network's lifetime while maintaining high trust accuracy.
- **Integration of Smart Contracts for Automated Trust Management:** Blockchain-based trust management uses smart contracts for automating key trust management processes, such as computing, validating, and disseminating trust scores. Through smart contracts, trust management can be enhanced and made more reliable, while also reducing the number of communications between nodes, thereby reducing network overhead.

## 1.3 Organization of the paper

The remainder of this paper is organized as follows. Sect 2 presents a comprehensive review of the state-of-the-art in trust management for VANETs, with a particular emphasis on blockchain-integrated approaches. This section identifies key research challenges and motivates the need for decentralized, tamper-proof trust frameworks. Sect 2 introduces the proposed blockchain-based decentralized trust management architecture. It outlines the system model, including the roles of On-Board Units (OBUs), Road Side Units (RSUs), and the Trust Authority (TA), and describes the mechanisms for vehicle authentication, trust evaluation, and selective message propagation.

Sect 3 details the technical foundation of the proposed framework, including the vehicle-to-vehicle authentication algorithm and the hybrid Proof-of-Stake (PoS) and Practical Byzantine Fault Tolerance (PBFT) consensus mechanism. The section formalizes trust computation strategies and block validation protocols under vehicular mobility constraints.

Sect 5 provides a thorough performance evaluation of the framework using simulations based on real-world VANET datasets. Key performance indicators such as trust accuracy, propagation latency, consensus overhead, scalability, and resilience against malicious behaviors are systematically analyzed.

Finally, Sect 7 concludes the paper by summarizing the contributions, emphasizing the significance of the proposed solution in addressing trust challenges in VANETs, and outlining potential directions for future work in the domain of secure vehicular communication systems.

## 2 Literature review

VANETs [14] rely on trust management to maintain secure and reliable communications between vehicles and infrastructure at the roadside [4]. A trust management system aims to prevent traffic accidents or inefficient network operations by assessing the trustworthiness of nodes in the network [19]. VANET trust management models can be broadly divided into two categories: centralized and distributed.

An entity called a trust manager is responsible for calculating and distributing trust scores in centralized trust management models [11]. Using this trust data, vehicles can assess the reliability of information they receive from other vehicles or roadside units. It offers a uniform and controlled way to manage trust across networks, but it also introduces significant challenges involving scalability, single points of failure, and delays in trust evaluation since it relies on a central server [17,26]. The model proposes a centralized trust management system that collects and processes trust-related data from all vehicles in the network. On the basis of vehicle behavior, historical interactions, and recommendations, the central authority calculates trust scores [27,31]. However, because this approach relies on a single entity for trust evaluation, the system is vulnerable to cyberattacks targeting the central authority, even though it ensures uniform trust evaluation across the network. With more vehicles on the road, scalability becomes a concern, which leads to processing bottlenecks and delays in trust computations.

In [24], the authors propose a centralized trust management framework that focuses on real-time data collection and processing through a cloud-based central trust server. The cloud infrastructure allows for high computational power, enabling more complex trust evaluation algorithms [30]. However, the use of cloud-based services introduces latency in trust score distribution, especially in dynamic VANET environments where vehicles frequently change locations. Additionally, the model dependence on continuous connectivity to the cloud server raises concerns regarding availability during network outages or poor connectivity.

In [20] introduces a reputation-based centralized trust model where the trust authority evaluates vehicles reputations based on feedback from other vehicles. The feedback is aggregated and weighted to calculate a reputation score for each vehicle. While the model effectively detects malicious vehicles, it suffers from long response times due to the feedback aggregation process, particularly in large-scale networks. The system is also susceptible to reputation manipulation attacks, where adversaries may collude to inflate or deflate trust scores.

A hierarchical centralized trust management model is proposed in [27], where the network is divided into zones, each managed by a local trust authority. These local authorities report to a global trust manager that oversees the entire network. While this model improves scalability by distributing trust management tasks across multiple entities, it introduces additional communication overhead between local and global trust authorities, leading to delays in trust dissemination. Furthermore, the global trust authority remains a central point of failure. In [32], a centralized trust management model is combined with machine learning techniques to predict trust scores based on vehicle behavior patterns. The model uses historical driving data and real-time behavioral analysis to assign trust scores. However, the high computational complexity of machine learning algorithms imposes significant demands on the central server, making the system difficult to scale for large VANET deployments. Additionally, the model is highly dependent on the availability of extensive historical data, limiting its applicability in scenarios where such data is not available.

The author in [19], explores a cloud-assisted centralized trust management model where a central cloud server evaluates trust based on vehicle sensor data and historical driving records. However, this reliance on cloud services introduces latency in trust score computation, particularly in areas with limited network coverage. Moreover, privacy concerns arise due to the central server's access to sensitive vehicle data. A centralized trust management model focusing on privacy-preserving techniques is proposed in [33], where the central authority computes trust scores without revealing vehicle identities. Although the model successfully protects vehicle privacy, the central server involvement in all trust-related transactions creates a bottleneck, resulting in performance degradation under heavy network loads. Furthermore, the

central server remains a potential target for privacy attacks, where adversaries could attempt to compromise the server to access sensitive information.

Distributed trust management models have gained significant attention in recent years due to their ability to scale and avoid single points of failure commonly associated with centralized models [32]. In distributed systems, vehicles autonomously compute and share trust information based on local observations and recommendations from neighboring nodes, allowing for decentralized trust establishment [34].

Distributed trust management models, in contrast, decentralize the process of trust evaluation by allowing individual vehicles to calculate trust scores based on local information, recommendations from neighboring vehicles, and other predefined criteria [7]. This decentralized approach enhances scalability and reduces the risk of single points of failure. However, distributed systems face challenges related to consistency and trust convergence, particularly in highly dynamic environments like VANETs, where vehicles frequently enter and leave the network [35]. In [1], the authors propose a distributed trust model in which each vehicle evaluates the trustworthiness of neighboring nodes based on direct interactions and the recommendations provided by other vehicles. The model improves scalability and reduces latency compared to centralized models by allowing trust to be computed locally. However, the accuracy of trust scores can be compromised in highly dynamic environments, where frequent vehicle movements disrupt the flow of recommendation data, leading to incomplete or inconsistent trust evaluations.

Recent literature presents several trust models for VANETs that do not utilize blockchain. For instance, Zhang et al. [39] proposed a reputation-based fuzzy logic system for detecting malicious behavior in VANETs. Similarly, Lee et al. [33] introduced a mobility-aware clustering trust mechanism using weighted behavior histories. While these systems offer low-latency decision-making and are computationally light, they often rely on centralized control units or semi-trusted RSUs, making them vulnerable to single-point failures and manipulation of trust metrics. Our approach addresses these limitations through blockchain distributed ledger, ensuring transparency, accountability, and tamper-resistance in trust evaluation. While these traditional trust systems are lightweight and often suitable for controlled VANET scenarios, they suffer from limited tamper-resistance and central-point vulnerabilities. Our blockchain-based trust framework addresses these shortcomings by introducing decentralized consensus, immutable logging, and dynamic validator selection. A comparative overview of key differences between non-blockchain and blockchain-based VANET trust systems is presented in Table 2, highlighting the advantages of our approach in terms of auditability, Sybil resistance, and decentralization.

Table 3 provides a comprehensive summary of blockchain-based trust management models for VANETs, highlighting methodologies, key findings, and limitations. The table outlines diverse approaches such as hybrid consensus mechanisms, privacy-preserving techniques, smart contract-based systems, and AI-enhanced trust evaluations. While these models offer improvements in trust accuracy, privacy, and scalability, challenges like computational overhead, energy consumption, and latency persist, emphasizing the need for balanced, efficient solutions in dynamic ITS environments. To facilitate a fair comparison, we standardized the evaluation metrics across different VANET trust models, as shown in Table 3. In particular, our proposed hybrid PoS-PBFT blockchain-based framework achieved the highest trust accuracy and low latency while maintaining scalability and robust security. This highlights the practical advantages of integrating blockchain in dynamic VANET environments.

**Table 2**. Comparison of Blockchain vs. Non-Blockchain trust mechanisms in VANETs.

| Aspect | Non-Blockchain | Blockchain-based (Ours) |
|---|---|---|
| Trust Storage | Local/RSU centralized | Decentralized ledger |
| Tamper Resistance | Low | High |
| Consensus | Not applicable/voting | PoS-PBFT hybrid |
| Sybil Resistance | Limited | Improved via trust-linked staking |
| Auditability | Limited | Full, on-chain traceable |

**Table 3**. Summary of blockchain-based trust management models in VANETs.

| Reference | Year | Methodology | Findings | Limitations |
|---|---|---|---|---|
| Liu et al. [24] | 2024 | Hybrid consensus (PoS & PBFT) for decentralized trust management | Improved security and reduced computational overhead | Scalability issues in large networks and stake accumulation attacks |
| Zhang et al. [22] | 2024 | Privacy-preserving trust management using zero-knowledge proofs | Enhanced privacy protection with secure identity verification | High computational overhead and delays in trust verification |
| Wang et al. [36] | 2024 | Smart contract-based reputation management with blockchain | Automated trust score calculation and secure data storage | High computational cost of smart contracts and energy consumption |
| Yang et al. [37] | 2024 | Lightweight consensus for efficient trust management | Significantly reduced computational burden and energy consumption | Reduced security guarantees compared to traditional consensus methods |
| Li et al. [33] | 2024 | PoW-based trust management model for secure VANET communication | Robust trust evaluation through decentralized validation | High energy consumption and computational delays |
| Patel et al. [38] | 2023 | Hybrid blockchain framework combining PoW and PBFT | Improved security and consensus speed for trust validation | Complexity in consensus process and increased latency |
| Wang et al. [36] | 2024 | Distributed blockchain-based reputation system for VANETs | Reduced reliance on centralized entities and enhanced privacy | Susceptibility to network congestion and delays in trust evaluation |
| Zang et al. [39] | 2024 | Federated blockchain-based trust management | Enhanced trust convergence through federated trust updates | Regional trust discrepancies and communication overhead |
| Huang et al. [40] | 2023 | Proof of Stake (PoS)-based trust management in VANETs | Low energy consumption and faster consensus process | Vulnerability to stake accumulation attacks and collusion |
| Liu et al. [6] | 2024 | Smart contract-based automated trust management | Automated trust score updates with tamper-proof records | High computational cost and delays due to smart contract execution |

The authors in [26] introduces a peer-to-peer (P2P) distributed trust management model where each node exchanges trust scores with its neighbors and aggregates the information to compute a final trust value. This model leverages the collaboration of nearby vehicles to establish trust without relying on a central entity. While the decentralized nature of the model enhances scalability and fault tolerance, it also increases the risk of trust manipulation attacks, such as collusion or Sybil attacks, where malicious nodes collaborate to artificially inflate trust scores.

A trust management model based on blockchain and distributed consensus is presented in [3], where vehicles participate in a consensus mechanism to validate trust information before it is propagated throughout the network. However, the integration of blockchain introduces computational overhead, particularly when consensus mechanisms such as Proof of Work (PoW) or Proof of Stake (PoS) are used, which can lead to delays in trust evaluation. In [41], a federated trust management model is proposed, in which vehicles locally compute trust scores and periodically share aggregated trust information with nearby roadside units (RSUs). The RSUs then validate and propagate the trust data to other regions. This approach reduces the communication overhead by minimizing the frequency of trust updates. A machine learning-enhanced distributed trust management model is explored in [42], where each vehicle uses reinforcement learning to dynamically adjust its trust evaluation based on the observed behavior of neighboring vehicles. The model is adaptive and self-learning, allowing vehicles to refine their trust scores over time.

## 2.1 Blockchain-based trust management models

Blockchain technology has recently gained traction as a potential solution for overcoming the limitations of both centralized and distributed trust management models in VANETs [18]. The decentralized, transparent, and immutable characteristics of blockchain make it well-suited for securing communication in highly dynamic and distributed environments like VANETs. In blockchain-based trust management models, trust scores and related information are recorded on a distributed ledger, ensuring that data is tamper-proof and verifiable by all participating nodes [11]. Blockchain technology has emerged as a promising solution to the limitations of both centralized and distributed trust management models. Each vehicle or roadside unit in the network can participate in the consensus process to validate trust data, ensuring that the trust scores are accurate and have not been manipulated. The decentralized nature of blockchain also removes the

reliance on a central trust manager, eliminating the risk of single points of failure and improving the overall security of the trust management system [43].

Several scholars have proposed blockchain-based trust management models specifically tailored for VANETs. In [44], a hybrid trust management model based on blockchain was proposed to combine vehicle-based trust calculations with blockchain consensus mechanisms. This approach ensures that trust scores are locally calculated and globally verified, offering a robust solution for secure communication in VANET environments. In [45], the authors propose a blockchain-based decentralized trust management system for VANETs that leverages a hybrid consensus mechanism combining Proof of Stake (PoS) and Practical Byzantine Fault Tolerance (PBFT). The system achieves high security and reduces computational overhead by utilizing PBFT for fast consensus and PoS to maintain decentralization. However, the hybrid model introduces challenges in terms of scalability, as the PoS mechanism requires nodes to validate transactions, which can become resource-intensive in large networks. [46] presents a blockchain-based trust management framework that focuses on privacy-preserving communication in VANETs. The model uses zero-knowledge proofs to verify vehicle identities without revealing personal information to the network. While this approach improves privacy and security, it introduces significant computational overhead during the verification process. The time delay caused by these cryptographic techniques can hinder real-time trust management in fast-moving vehicular networks. In [47] proposes a lightweight blockchain-based trust management system designed to reduce the computational burden typically associated with blockchain technology. The model integrates a lightweight consensus algorithm that minimizes the energy consumption and latency of blockchain operations. Although the model significantly improves computational efficiency, it does so at the cost of reduced security guarantees compared to more robust consensus mechanisms like Proof of Work (PoW) or PoS. The lightweight consensus may not be resilient against sophisticated attacks such as double-spending or Sybil attacks, particularly in highly dynamic environments like VANETs [27].

In [48], the authors implement a blockchain-based trust management model that combines vehicle-to-vehicle (V2V) communication with blockchain-based trust validation. Each vehicle calculates a local trust score and periodically submits the score to the blockchain, where it is validated through a consensus process. While this model enhances the scalability of the trust management process by distributing the computational load across multiple vehicles, it introduces communication overhead due to frequent interactions between vehicles and the blockchain network. The increased communication can lead to network congestion, particularly in dense traffic scenarios, thus reducing overall system performance.

## 3 Methodology

This section presents a decentralized trust evaluation mechanism for VANETs. This mechanism incorporates selective data propagation and a decentralized consensus mechanism to ensure efficient trust management while minimizing network overhead and ensuring scalability in highly dynamic vehicular environments.

### 3.1 System components

The proposed system architecture consists of three main components: On-Board Units (OBUs) installed in vehicles, Roadside Units (RSUs) deployed along transportation infrastructure, and a Trust Authority (*TA*) that oversees network initialization and credential issuance. The *TA* is responsible for registering entities, issuing cryptographic credentials, and setting global system parameters to ensure secure and verifiable identities prior to vehicles joining the network.

Importantly, the blockchain network, not the Trust Authority (*TA*), is responsible for decentralized trust evaluation, consensus, and ledger maintenance. The *TA* does not participate in operational trust computation or block validation; these functions are fully carried out by distributed validator nodes using the hybrid PoS PBFT mechanism. This separation of responsibilities ensures that while the *TA* provides regulatory oversight and initialization, the trust management process itself remains fully decentralized and tamper resistant.

Vehicles communicate through Vehicle-to-Vehicle (V2V) and Vehicle-to-Infrastructure (V2I) links to exchange information and generate trust scores based on direct and indirect interactions. All validated data are recorded on the blockchain ledger, enabling transparent, auditable, and secure trust management without centralized control.

**3.1.1 Vehicles (On-Board Units - OBU).** Each vehicle in the VANET is equipped with an On-Board Unit (OBU) that facilitates communication with other vehicles (Vehicle-to-Vehicle, V2V) and infrastructure (Vehicle-to-Infrastructure, V2I). The OBU is responsible for computing trust scores based on interactions with neighboring vehicles. Trust evaluations are made locally by each vehicle, considering both direct interactions and reputational feedback from nearby nodes.

**3.1.2 Roadside Units (RSUs).** RSUs are fixed infrastructure units located along roads that facilitate communication between vehicles and the blockchain network. They collect trust-related data from vehicles and forward it for validation through the consensus mechanism. RSUs help manage trust updates and ensure that critical data is shared between vehicles and the broader network.

## 3.2 Trust Authority (TA)

The Trust Authority (*TA*) plays a critical supervisory and bootstrapping role in our decentralized trust management system. While the *TA* is assumed to be secure and reliable, as in similar works [33], it does not act as a centralized controller. Instead, it facilitates decentralized trust operations in VANETs by overseeing credential issuance, validator selection, and trust bootstrap processes. In our architecture, the *TA* assists in initializing the network by securely registering OBUs and RSUs, issuing cryptographic credentials, and managing the setup of validator nodes. Once initialization is complete, trust evaluations, block generation, and consensus operations are entirely decentralized and carried out through validator nodes using the hybrid PoS–PBFT mechanism.

Furthermore, the *TA* maintains visibility over vehicle data and supports real-time identification of potentially malicious entities [49]. It collects the trust scores, $T(v_i, t)$, for each vehicle $v_i$ over time $t$, where the scores are derived from both direct and indirect interactions as follows:

$$T(v_i, t) = \alpha \cdot D(v_i, t) + \beta \cdot \sum_{j=1}^{n} w_j R(v_i, v_j, t),$$ (1)

where $D(v_i, t)$ represents the direct trust score based on prior behavior, and $R(v_i, v_j, t)$ denotes the recommendation of node $v_j$ regarding $v_i$. Each recommendation is weighted by $w_j$, which reflects the recommender's trust level and is computed as a normalized trust value over recent interactions:

$$w_j = \frac{T(v_j, t - \Delta t)}{\sum_{k=1}^{n} T(v_k, t - \Delta t)}.$$ (2)

The parameters $\alpha$ and $\beta$ are weighting coefficients with $\alpha + \beta = 1$, and $n$ is the number of neighboring nodes. For newly joined vehicles, the *TA* provides an initial trust value $T_0(v_i)$ to establish a starting reputation profile. This initialization ensures that new participants can engage in the network immediately while their trust scores are continuously updated through decentralized evaluation and consensus as they interact with other nodes.

As vehicular dynamics evolve, trust updates remain responsive yet stable through exponentially weighted moving averages (EWMA) in both $D(v_i, t)$ and $R(v_i, v_j, t)$. Importantly, these initialization and monitoring processes do not interfere with or influence consensus decisions. All trust updates are recorded on-chain and validated through the distributed validator network.

The rationale for incorporating a *TA* is grounded in practical deployment scenarios, where regulatory or transportation authorities require auditability and accountability in safety-critical networks like VANETs. The *TA* may issue revocations and compliance notices; however, it possesses no decision-making authority over consensus or trust score updates. Even

if the *TA* were compromised, it could not retroactively alter trust records or control consensus, since these functions are governed by the blockchain network.

**3.2.1 Convergence analysis and TRF threshold selection.** This subsection provides (i) a brief convergence analysis for the EWMA updates used in direct and recommendation-based trust components, and (ii) a principled procedure for selecting the Trust Relevance Factor (TRF) threshold used in selective propagation.

Let the direct-trust update follow the EWMA recursion

$$D_t = (1 - \lambda)D_{t-1} + \lambda I_t, \tag{3}$$

where $0 < \lambda \leq 1$ is the smoothing parameter and $I_t$ is the interaction outcome at time $t$. Assume $\{I_t\}$ is a stationary sequence with mean $\mu_I$ and variance $\sigma_I^2$. Taking expectations yields

$$\mathbb{E}[D_t] = (1 - \lambda)^t D_0 + \mu_I(1 - (1 - \lambda)^t), \tag{4}$$

so that,

$$\lim_{t \to \infty} \mathbb{E}[D_t] = \mu_I.$$

For the variance, assuming independence between innovation terms for simplicity, the steady-state variance of $D_t$ is

$$\lim_{t \to \infty} \text{Var}(D_t) = \frac{\lambda \sigma_I^2}{2 - \lambda}. \tag{5}$$

Thus the EWMA estimator is unbiased in the limit and its variance is bounded and controlled by $\lambda$. Smaller $\lambda$ yields smoother but slower adaptation; larger $\lambda$ increases responsiveness at the cost of higher steady-state variance.

**3.2.1.1 Boundedness and convergence of the composite trust score.** The composite trust score is given by

$$T_t = \alpha D_t + \beta \sum_{j=1}^{n} w_{j,t} R_{j,t}, \tag{6}$$

with $\alpha, \beta \geq 0$, $\alpha + \beta = 1$, and normalized weights $w_{j,t} \geq 0$ satisfying $\sum_{j=1}^{n} w_{j,t} = 1$. If each component $D_t$ and $R_{j,t}$ is bounded in [0,1], then $T_t \in [0, 1]$ for all $t$. Under mild ergodicity conditions on the underlying interaction/recommendation processes, each component converges in expectation to its long-run mean and therefore $T_t$ converges in expectation to a convex combination of these means. Because the weights $w_{j,t}$ are normalized and derived from past trust values, they remain bounded and do not induce divergence.

**3.2.1.2 Definition and normalization of TRF.** Recall TRF between nodes $v_i$ and $v_j$ is defined as

$$\text{TRF}_{ij} = \gamma F_{ij} + \delta I_{ij}, \tag{7}$$

where $F_{ij}$ is the interaction frequency (normalized to [0,1] over a sliding window) and $I_{ij}$ is the importance metric (also normalized to [0,1]). For clarity and stability we recommend imposing $\gamma, \delta \geq 0$ and $\gamma + \delta = 1$; normalization of $F_{ij}$ and $I_{ij}$ should be performed by min–max or by dividing by the maximum observed value over the calibration window.

**3.2.1.3 Principled TRF threshold selection.** We recommend a hybrid, reproducible approach comprising three options depending on the availability of labeled calibration data and deployment constraints.

**Option A (ROC-based calibration, preferred when labeled calibration data are available).** Let $\tau$ denote the TRF threshold and define the detection rule "propagate if $\text{TRF}_{ij} > \tau$". Compute true positive rate $\text{TPR}(\tau)$ and false positive rate $\text{FPR}(\tau)$ over a labeled calibration set. Choose

$$\tau^* = \arg\max_{\tau}\left(\text{TPR}(\tau) - \text{FPR}(\tau)\right), \tag{8}$$

i.e., maximize Youden's J statistic to obtain the operating point with the best trade-off between sensitivity and specificity. Alternatively, select $\tau$ to satisfy a deployment-specific constraint on FPR.

**Option B (statistical adaptive rule, for online operation).** Maintain a rolling window of the most recent $W$ TRF samples among relevant neighbors, compute the rolling mean $\mu_{\text{TRF}}$ and standard deviation $\sigma_{\text{TRF}}$, and set

$$\tau(t) = \mu_{\text{TRF}}(t) + k\,\sigma_{\text{TRF}}(t), \tag{9}$$

where $k$ is chosen to control the Type I error (e.g., $k = 1.96$ for a one-sided 97.5% threshold under approximate normality). This rule adapts to local traffic density and behavior.

**Option C (percentile-based fallback, for deployments without labeled data).** Compute the empirical $p$-th percentile $Q_p$ of TRF over a calibration period and set $\tau = Q_p$ with $p$ in [0.70, 0.90] depending on desired sensitivity.

For typical vehicular network deployments, we recommend the following initial parameter settings: an EWMA smoothing factor $\lambda \in [0.05, 0.25]$, where smaller values yield smoother but slower trust updates; a TRF importance weight $\delta \in [0.5, 0.7]$, with $\delta = 0.6$ used in our baseline experiments to emphasize message criticality; and a calibration window $W$ chosen based on expected neighbor churn. These parameters should subsequently be validated and fine-tuned using real deployment traces or the VeReMi dataset through the ROC-based calibration procedure described above.

### 3.3 Blockchain-enabled trust management and selective propagation

The proposed VANET trust management framework leverages blockchain to ensure a decentralized, secure, and tamper-resistant infrastructure for trust evaluation and enforcement. At the core of this architecture, each vehicle is equipped with an On-Board Unit (OBU), enabling real-time communication with other vehicles (V2V) and Roadside Units (RSUs) through Vehicle-to-Infrastructure (V2I) communication. RSUs act as relay nodes that assist in transmitting trust-related data to the blockchain network, forming the backbone of secure vehicular communication. Fig 2 illustrates the overall architecture, showcasing the interplay between OBUs, RSUs, the TA, and the blockchain infrastructure. Vehicles exchange trust-relevant information during mobility, which is then filtered, validated, and securely stored in the blockchain. Smart contracts enforce automatic trust updates and policies, ensuring system autonomy and eliminating the need for centralized control. The proposed system integrates vehicular communication, trust evaluation, consensus validation, blockchain-based recordkeeping, and selective data dissemination to provide a robust, decentralized trust management framework tailored for VANET environments.

Trust computation is coordinated by a semi-centralized Trust Authority (TA), which plays a supervisory but non-authoritative role. It aggregates direct interaction outcomes and peer feedback to generate a dynamic trust score $T(v_i, t)$ for each vehicle, as formalized in Eq (1). The TA also assists in identifying misbehaving entities, ensuring that the reputation system remains adaptive and responsive. However, the trust evaluation outcomes are not finalized until they pass through a decentralized consensus process, ensuring resilience against central point failures. The trust scores and validated vehicular interactions are transmitted to a blockchain layer, where they are permanently recorded in a transparent, immutable ledger. This blockchain ledger serves as the system's trust repository, storing all verified transactions in sequential blocks (Block 1, Block 2, etc.), ensuring auditability and historical traceability. These blocks form a cryptographically linked chain that supports tamper-proof recordkeeping of vehicular trust histories. Consensus on the

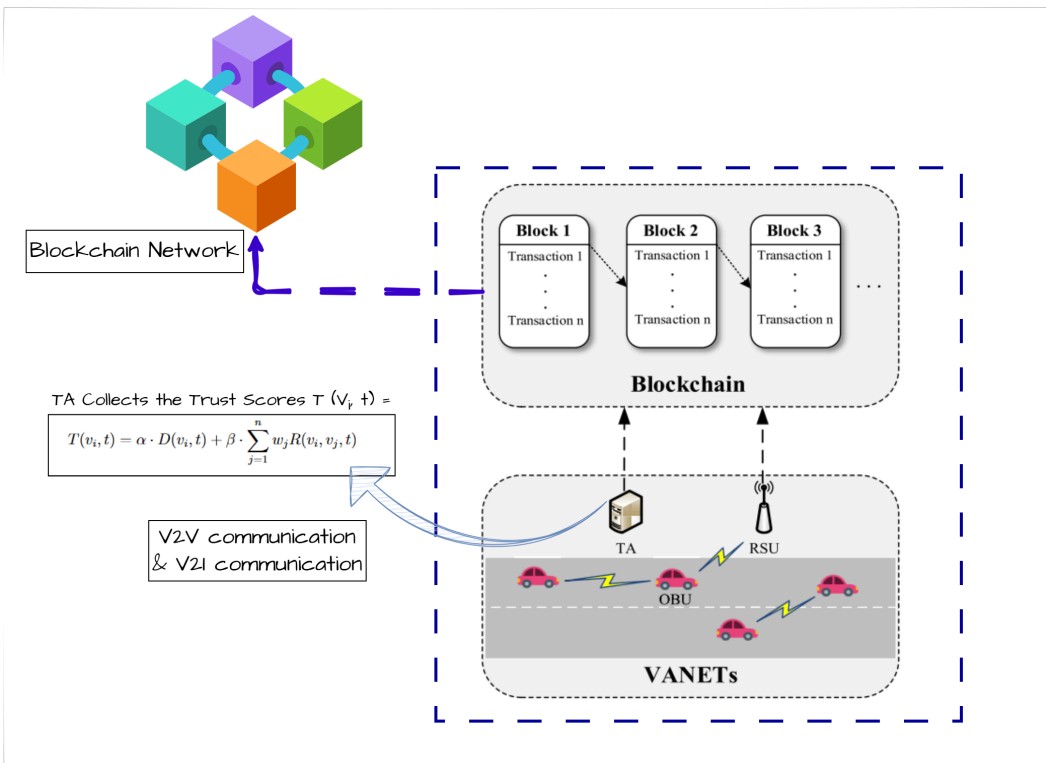

**Fig 2**. **Proposed blockchain-based VANET system model.** The architecture consists of On-Board Units (OBUs), Roadside Units (RSUs), and a Trust Authority (TA) with a bootstrapping and supervisory role for credential issuance, identity registration, and validator setup. The TA does not participate in trust computation, consensus, or ledger maintenance. Vehicles communicate via V2V and V2I links, while trust scores $T(v_i, t)$ are derived from direct interactions and peer recommendations. All validated data are stored on a tamper-proof blockchain maintained by distributed validators, ensuring decentralized, auditable, and secure trust management with regulatory oversight.

validity of transactions is achieved through a hybrid Proof of Stake (PoS) and Practical Byzantine Fault Tolerance (PBFT) mechanism. Consensus validators are selected based on accumulated trust stakes, promoting fairness and energy efficiency through PoS, while PBFT ensures fault tolerance and rapid finality by requiring agreement among a majority of trustworthy nodes. This hybrid consensus framework maintains high security and performance, making it suitable for high-mobility vehicular environments.

To avoid overwhelming the network with redundant updates, a selective data propagation mechanism is incorporated. Rather than broadcasting all trust-related data globally, only contextually relevant updates are disseminated to nodes that have had recent or meaningful interactions with the evaluated vehicle [50]. This targeted approach significantly reduces communication overhead and enhances system scalability without compromising trust transparency.

### 3.4 Blockchain-based trust management scheme

This section presents our proposed trust management scheme in detail, which comprises four parts, namely, system initialization, vehicle credibility evaluation, event credibility evaluation, and storage of indirect trust through blockchain. The comprehensive explanation of our suggested scheme can be stated as follows.

## 3.5 System initialization

- *TA* defines an elliptic curve $E : y^2 \equiv x^3 + ax + b \pmod{q}$, where $G$ is a finite cyclic group of order $q$ on the elliptic curve, $q$ is a large prime number, $P$ is the generator of the group $G$, and $a, b \in \mathbb{Z}_q^*$, with $\mathbb{Z}_q^* = \{1, 2, \ldots, q-1\}$. Then, *TA* exposes these parameters to all nodes.
- To join the network, vehicle $V_i$ forwards a registration request to the RSU, generating a random number $S_{ki}$ as its private key and producing its public key $P_{ki} = P \cdot S_{ki}$. $V_i$ further generates its certificate Cert$_i$ utilizing the private key, which is then transmitted along with the public key to the RSU.
- Upon receiving the registration request, the RSU stores the vehicle's certificate Cert$_i$ and the indirect trust $IT_i = 0$ on the blockchain. The RSU deploys smart contracts and utilizes the vehicle's public key as an identifier to retrieve certificates and trust values, reducing the storage burden on the vehicle. It is assumed that the RSU entities are trustworthy. Additionally, Bayesian classifiers trained on the VeReMi dataset are used to enable vehicles to compute the direct trust of the target vehicle.

## 3.6 Evaluation of vehicle credibility

Evaluating the credibility of vehicles is a crucial part of the trust management process. This assessment involves calculating the vehicle trust value by combining direct and indirect trust. Direct trust refers to the credibility established through direct observations or interactions, while indirect trust is derived from evaluations made by other vehicles.

**1) Direct trust calculation**: Direct trust is calculated using a GaussianNB classifier, which is trained on the refined VeReMi dataset. For each vehicle, its state information $S$ is input to the Bayesian classifier, which computes the direct trust $DT_i$. The specific process for computing the direct trust will be discussed in the "Trust Value Calculation" section.

**2) Indirect trust calculation:** The indirect trust $IT_i$ of a newly registered vehicle is initialized to zero. Our method employs an active detection scheme where neighboring vehicles and the RSU work collaboratively to detect the trust level of the target vehicle, as shown in Eq 10. The neighbor vehicle $V_j$ sends a test message MSG $= \{M, \text{Sig}, P_{kj}, \text{Tsp}\}$ to vehicle $V_i$, requesting it to forward the message to the RSU. The RSU compares the message received from $V_i$ and $V_j$. If both are identical, $V_i$ is considered legitimate, and a trust score $T_i = 1$ is assigned. If not, $V_i$ is deemed malicious, and a score of $T_i = 0$ is sent to the TA. The indirect trust $IT_i$ is computed as:

$$IT_i = \frac{1}{n} \sum_{j=1}^{n} T_i(j) \tag{10}$$

**3) Updating Indirect Trust:** For already registered vehicles, the indirect trust $IT_i$ is updated using the formula:

$$IT_i^{\text{new}} = (1 - \theta_1) IT_i^{\text{old}} + \theta_1 T_i \tag{11}$$

Here in Eq 11, $\theta_1$ is a weight parameter that controls the rate of change in indirect trust. This allows for dynamic adjustment during the detection of malicious vehicles. The updated trust values are stored on the blockchain via the RSU using a consensus algorithm.

**4) Trust Value Calculation:** The overall trust value Trust$_i$ for a vehicle is calculated by combining direct trust $DT_i$ and indirect trust $IT_i$ is computed in Eq 12:

$$\text{Trust}_i = \frac{DT_i + IT_i}{2} \tag{12}$$

The trust value must exceed a threshold $\delta$ for the vehicle to be considered credible.

### 3.7 Evaluation of event credibility

In addition to vehicle trust, assessing the credibility of events is essential to prevent the dissemination of false information, which can lead to serious consequences in VANETs [51]. The event credibility evaluation process ensures that the events reported by vehicles are verified before being accepted and broadcasted to the network. The steps involved in this process are described below.

**1) Event Reporting by Vehicle:** When a vehicle $V_i$ detects an event $E$ such as a traffic accident or hazard, it broadcasts the event information to surrounding vehicles $V_j = \{V_1, V_2, \dots, V_n\}$ and requests these vehicles to provide their judgment on the event.

**2) Evaluation Accuracy Calculation:** The accuracy of the information provided by the vehicle can vary depending on several factors, including the trust level of the vehicle and its distance from the event location. To account for these factors, each surrounding vehicle $V_j$ computes its evaluation accuracy $C_j$ using the following formula:

$$C_j = \text{Trust}_j \cdot e^{-\theta_2 d_j} \tag{13}$$

Here in Eq 13, $\text{Trust}_j$ is the trust value of vehicle $V_j$, $d_j$ is the distance between vehicle $V_j$ and the location of the event $E$, $e$ is Euler constant, and $\theta_2$ is a weight parameter that adjusts the impact of the distance on the evaluation accuracy. By adjusting $\theta_2$, the system can modify the degree to which the distance affects the credibility of the event.

Fig 3 presents a blockchain-based model for event credibility evaluation in a VANET system. When an event occurs, such as an accident or hazard, a vehicle $V_i$ initiates a message broadcast with relevant event data, including context $C_i$ and transmission rate $\text{Rate}^i$. Nearby vehicles, represented as the set $V_j = \{V_1, V_2, \dots, V_n\}$, receive the message and send verification requests to the Road Side Unit (RSU). The RSU aggregates verification requests and calculates credibility scores based on factors such as vehicle context and message integrity, formalized as $\text{Rate}^j = f(V_j, C_j)$. The RSU then

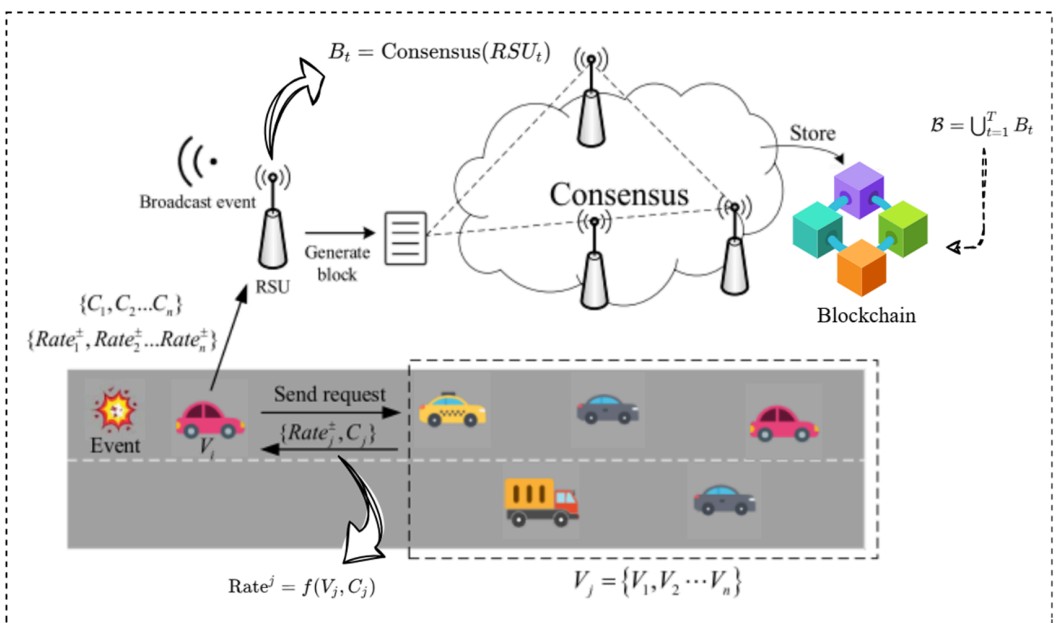

**Fig 3**. **Blockchain-based model for event credibility evaluation in VANETs, utilizing RSU consensus to secure and verify event data on the blockchain.**

initiates a consensus process, denoted as $B_t = \text{Consensus(RSU)}$, where multiple RSUs collaborate to confirm the event's credibility. The consensus results are recorded in a new block $B_t$ that is subsequently stored in the blockchain ledger $B = \bigcup_{t=1}^{T} B_t$, ensuring that verified event data remains immutable and tamper-proof. This model enhances data integrity and trust in VANET communications by securely evaluating event credibility.

**3) Judgment of Event:** After calculating its evaluation accuracy, each vehicle $V_j$ provides its judgment of the event, denoted as $\text{Rate}_j^+ = 1$ for a positive judgment (indicating that the event is valid) or $\text{Rate}_j^- = -1$ for a negative judgment (indicating that the event is invalid). These judgments are then sent back to the vehicle $V_i$ that initially reported the event.

**4) Calculation of Event Probability by RSU:** Vehicle $V_i$ aggregates the judgments from surrounding vehicles and forwards the data to the nearest RSU. The RSU calculates the probability of event $E$ being true using the following equation:

$$P(E) = \frac{\sum_{j=1}^{n} C_j \cdot \text{Rate}_j^+}{\sum_{j=1}^{n} C_j \cdot \text{Rate}_j^+ - \sum_{j=1}^{m} C_j \cdot \text{Rate}_j^-} \tag{14}$$

In Eq 14, $n$ represents the number of positive judgments received for the event, and $m$ represents the number of negative judgments. The probability $P(E)$ determines the likelihood of the event being credible based on the aggregated evaluations.

**5) Broadcasting the Event:** If the probability $P(E)$ exceeds a predefined threshold, the event $E$ is deemed credible and the RSU broadcasts the event to all vehicles in the network. The indirect trust of the participating vehicles is updated accordingly. Conversely, if $P(E)$ falls below the threshold, the event is considered not credible, and the RSU refrains from broadcasting it. Vehicles that falsely reported the event may be penalized by reducing their trust value. The formulas for updating indirect trust are as follows:

$$\text{IT}_j^{\text{new}} = (1 - \theta_3)\text{IT}_j^{\text{old}} + \theta_3 \cdot \text{Rate}_j \tag{15}$$

Eqs 15 and 16, $\theta_3$ is a weight parameter that controls the rate of trust adjustment based on the vehicle judgment. For the vehicle that initially reported the event, the indirect trust is updated as:

$$\text{IT}_i^{\text{new}} = \frac{\text{IT}_i^{\text{old}} + P(E)}{2} \tag{16}$$

These updates ensure that vehicles contributing accurate judgments maintain or improve their trust levels, while those providing inaccurate information experience a decrease in trust.

## 3.8 Decentralized trust evaluation

The proposed model evaluates trust in a decentralized manner, where each node independently calculates the trust score of its neighboring nodes based on both direct interactions and recommendations from other nodes. The total trust score $T(v_i)$ of a node $v_i$ at time $t$ is defined as a weighted combination of direct trust and reputation-based trust [52]:

$$T(v_i, t) = \alpha D(v_i, t) + \beta \sum_{j=1}^{n} w_j R(v_i, v_j, t) \tag{17}$$

where in Eq 17:

- $D(v_i, t)$ is the direct trust score of node $v_i$ at time $t$, computed based on its interactions.
- $R(v_i, v_j, t)$ represents the recommendation provided by node $v_j$ regarding $v_i$ at time $t$.

- $n$ is the number of neighboring nodes that provide recommendations.
- $w_j$ is the weight assigned to node $v_j$'s recommendation based on its historical reliability.
- $\alpha$ and $\beta$ are weights that balance the contribution of direct trust and reputational trust, such that $\alpha + \beta = 1$.

The direct trust $D(v_i, t)$ is calculated based on node $v_i$ behavior during interactions, such as message authenticity, consistency, and adherence to protocol. It is updated over time based on exponential moving averages (EMA) of past interactions, providing a smoothed trust evaluation:

$$D(v_i, t) = (1 - \lambda) \cdot D(v_i, t - 1) + \lambda \cdot I(v_i, t) \tag{18}$$

where in Eq 18:

- $I(v_i, t)$ represents the interaction outcome at time $t$ (i.e., a binary value indicating whether the interaction was trustworthy or not).
- $\lambda$ is a smoothing factor that controls the influence of recent interactions over past ones.

In this decentralized model, each node autonomously updates its trust evaluations based on its observations and shared reputations, without relying on any centralized authority. This ensures scalability and resilience in dynamic network environments.

### 3.9 Selective data propagation

To address the communication overhead and scalability challenges inherent in VANETs, we introduce a selective data propagation mechanism driven by a Trust Relevance Factor (TRF). Unlike traditional gossip-based dissemination protocols which often suffer from excessive redundancy and delayed convergence in highly dynamic networks. Our approach employs a deterministic, relevance-aware strategy that selectively targets only those nodes with meaningful interaction histories.

The core idea hinges on the TRF metric, which quantifies the necessity of propagating trust updates based on both the frequency and criticality of interactions between nodes. Specifically, the TRF between nodes $v_i$ and $v_j$ is defined as follows:

$$TRF(v_i, v_j) = \gamma \cdot F(v_i, v_j) + \delta \cdot I(v_i, v_j) \tag{19}$$

Here, $F(v_i, v_j)$ denotes the frequency of interactions within a sliding time window $w$, calculated as:

$$F(v_i, v_j) = \frac{\text{interactions}(v_i, v_j)}{w}$$

while $I(v_i, v_j)$ captures the importance or criticality of exchanged information (e.g., safety or event-based messages). The weighting parameters $\gamma$ and $\delta$ allow the system to tune the influence of interaction frequency and importance, respectively. Through empirical tuning, we set $\delta = 0.6$ to balance detection sensitivity and false positives under mobility-induced uncertainty.

When the TRF exceeds a predefined threshold $\tau$, a trust update is disseminated as:

$$\text{Propagate if } TRF(v_i, v_j) > \tau \tag{20}$$

To identify relevant peers for propagation, each vehicle maintains a dynamically updated neighbor table constructed from periodic beaconing specifically Basic Safety Messages (BSMs), standard in DSRC and C-V2X protocols. This

passive discovery mechanism enables vehicles to track neighboring nodes in real-time and determine which among them require trust updates based on the TRF computation. The process is inherently localized, eliminating the need for network-wide broadcasts or global state awareness. By disseminating trust information solely to impacted neighbors, the proposed scheme significantly reduces redundant communication while ensuring that vital trust updates reach the appropriate nodes. This localized propagation approach ensures bandwidth efficiency and preserves system responsiveness even in densely populated or high-mobility vehicular environments. Moreover, in contrast to probabilistic gossip-based protocols, our TRF-guided approach is deterministic, yielding better convergence and lower communication overhead. As demonstrated in Sect 5, our mechanism achieves over 95% trust accuracy while reducing communication overhead by approximately 35%, outperforming conventional epidemic dissemination models.

Lastly, the proposed scheme operates within a decentralized trust management framework, validated using a hybrid consensus mechanism that combines Proof of Stake (PoS) and Practical Byzantine Fault Tolerance (PBFT). This ensures both the integrity and consistency of trust scores across the network, supporting secure and scalable trust validation without reliance on centralized infrastructure.

### 3.10 Enhanced hybrid consensus mechanism for VANETs: Mobility-aware PoS with adaptive PBFT

In dynamic vehicular environments, directly applying conventional consensus protocols such as PoS or PBFT faces fundamental challenges related to validator volatility, stake stability, and consensus liveness. To address these issues, we design a hybrid consensus mechanism that combines mobility-aware Proof of Stake (PoS) with adaptive Practical Byzantine Fault Tolerance (PBFT). Unlike previous approaches that simply merge these protocols, our design introduces validator selection and consensus adaptations tailored specifically for VANET mobility patterns.

**3.10.1 Mobility-aware validator selection via PoS.** Traditional PoS mechanisms rely on static, coin-based stake accumulation, which is incompatible with the highly mobile and transient nature of vehicular networks. To overcome this, we define a dynamic trust stake $S(v_i, t)$ that reflects the recent behavior and trustworthiness of each node:

$$S(v_i, t) = T(v_i, t) = \alpha \cdot D(v_i, t) + \beta \cdot \sum_{j=1}^{n} w_j R(v_i, v_j, t), \tag{21}$$

where $D(v_i, t)$ represents direct trust and $R(v_i, v_j, t)$ denotes recommendations from neighboring nodes. Stake values are computed over a sliding time window, ensuring that only nodes with consistently trustworthy behavior are eligible for validator selection. Validators are chosen from a short-term validator pool composed of RSUs and low-mobility OBUs within a geofenced area. The validator selection probability is given by:

$$P(v_i) = \frac{S(v_i, t)}{\sum_{j=1}^{n} S(v_j, t)}. \tag{22}$$

Re-selection occurs periodically (every 5–10 seconds), preventing transient nodes from accumulating stake over long periods and maintaining validator stability during high-speed mobility. This mobility-aware design effectively addresses the stake persistence issue in traditional PoS.

**3.10.2 Adaptive PBFT with geofenced validator sets.** Once the validator pool is formed, PBFT is executed within localized geofenced groups to maintain a stable validator set over short intervals. We introduce adaptive quorum tuning and rapid view-change mechanisms to handle validator dropouts. If a validator leaves the geofenced area during consensus, the protocol triggers an immediate view change, electing a new primary from the remaining validators using cached stake values.

Consensus proceeds through the standard PBFT phases (pre-prepare, prepare, commit), but quorum thresholds are dynamically adjusted based on the active validator set size to maintain liveness and safety under mobility. Redundant state synchronization ensures consistency even during validator transitions.

**3.10.3 Mobility-aware validator transition handling.**  A key novelty of our approach lies in explicitly handling validator transitions under mobility. Validator sets are bounded within geofences, and stake information is periodically cached at RSUs. When a validator moves out of range, RSUs facilitate a seamless handover by initiating view change and re-selection within the same consensus round, avoiding stalls or consensus restarts. This mechanism enables the hybrid protocol to maintain low latency and deterministic validator transitions, which traditional PBFT cannot achieve in highly dynamic environments.

### 3.11 Storage of indirect trust through blockchain

The storage of vehicle certificates and indirect trust values on a blockchain is proposed to prevent malicious tampering of data and ensure secure storage.

Blockchain decentralized and immutable nature significantly enhances system security. However, the delay in updating trust values depends on the consensus algorithm utilized by the blockchain. The PBFT algorithm, though effective for networks with a smaller number of nodes, faces scalability issues as it requires pairwise communication among all nodes [38]. As the number of participating nodes increases, the communication overhead becomes prohibitive.

As shown in Fig 4, the hybrid consensus mechanism combines PoS for dynamic selection of validators and PBFT for achieving agreement on trust scores. The system recalculates PoS-based trust stakes periodically to account for the mobility of vehicles in VANETs, while PBFT rounds ensure low-latency, secure finalization of blocks. This approach balances scalability, security, and efficiency in managing indirect trust on the blockchain [38].

Our system adopts the BFT-Delegated Proof of Stake (BFT-DPoS) consensus algorithm, which offers a more scalable alternative. In this approach, representative nodes are selected through a voting process to handle the validation and verification tasks. Upon receiving a notification of a new block, these representative nodes verify the integrity and legitimacy

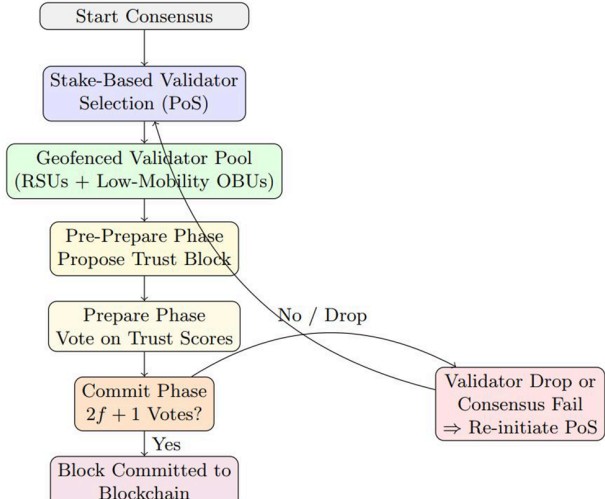

**Fig 4**. **Flow of the proposed hybrid consensus mechanism integrating Proof of Stake (PoS) for dynamic validator selection and Practical Byzantine Fault Tolerance (PBFT) for trust score agreement.** The system adapts to VANET mobility by periodically recalculating PoS-based trust stakes and using low-latency PBFT rounds to finalize blocks securely and efficiently.

of the block's transactions. Once a majority of the representative nodes confirm the block's validity, it is appended to the blockchain.

The BFT-DPoS algorithm operates as follows: The system elects 21 nodes with the highest votes as super nodes. These super nodes then select a miner among themselves based on network resources. The miner broadcasts a signed request with a timestamp to all super nodes. Upon verifying the request, the super nodes audit the results and communicate their findings to other nodes. When the audit results from the super nodes are consistent, and the number of matching results exceeds $2f$, where $f$ represents the number of Byzantine nodes, a confirmation message is broadcast to all nodes. If the super node collects more than $2f + 1$ confirmations, the block is validated and added to the blockchain. If consensus is not reached, the miner initiates additional consensus rounds to resolve any discrepancies.

## 3.12 Network overhead minimization

The combination of selective data propagation and decentralized consensus significantly reduces network overhead. First, by limiting trust updates to only relevant nodes, the model minimizes unnecessary communication and bandwidth usage. Second, the PoS mechanism ensures that only a subset of nodes participate in the consensus, reducing the computational burden on the network. The total communication overhead $O_{comm}$ can be modeled as:

$$O_{comm} = O_{local} + O_{propagate} \tag{23}$$

where $O_{local}$ represents the overhead for local trust updates, and $O_{propagate}$ represents the overhead for propagating trust data to neighboring nodes when $TRF > \tau$, as shown in Eq 23. By controlling $O_{propagate}$ through selective propagation, the model ensures that communication costs remain low.

The proposed methodology provides a decentralized and efficient approach to trust management in VANETs. By using selective data propagation and decentralized consensus, the model achieves secure and scalable trust evaluations while minimizing network overhead. The mathematical formulation of trust evaluation, relevance factor, and consensus ensures that the system is both robust and computationally efficient.

## 3.13 Comparison with traditional PKI-based authentication

Traditional Public Key Infrastructure (PKI) mechanisms rely on certificate authorities (CAs) to validate the identity of nodes in vehicular networks. These systems offer low-latency cryptographic authentication and are effective in relatively stable environments. However, VANETs operate in dynamic, large-scale, and often adversarial conditions, where static certificate verification is insufficient to assess the ongoing behavior of participating vehicles.

In contrast, our proposed blockchain-based mechanism integrates identity verification with dynamic trust evaluation by invoking smart contracts. While this may introduce marginal latency due to blockchain queries, it ensures that both the identity and historical behavior (trust profile) of the sender vehicle are jointly evaluated. This dual-layer verification enhances security and context awareness, particularly in detecting on-off attackers and sybil nodes that may possess valid certificates but exhibit malicious behavior. Moreover, anchoring trust evaluations on-chain offers several advantages: (i) it enables tamper-resistant and auditable logging of trust decisions; (ii) it supports cross-domain trust interoperability without reliance on a centralized authority; and (iii) it facilitates decentralized revocation and trust updates in real-time. Lightweight query mechanisms and local caching at RSUs further minimize overhead and ensure scalability in practical deployments.

Therefore, while PKI mechanisms excel in static identity management, the proposed blockchain-enabled trust-aware authentication better aligns with the dynamic and adversarial nature of VANETs, offering a robust, decentralized alternative that goes beyond identity to include behavioral validation.

## 4 Smart contract design

The proposed authentication mechanism, illustrated in Algorithm 1, leverages smart contracts to retrieve identity tags and prior trust evaluations of communicating vehicles. This design provides a holistic assessment of a vehicle trustworthiness by combining cryptographic identity verification with on-chain behavioral history. A detailed comparison with conventional PKI-based mechanisms is provided in Sect 3.13. To achieve secure, fault-tolerant, and efficient consensus on trust values within VANETs, we propose a hybrid consensus mechanism that integrates PoS and PBFT, as presented in Algorithm 2. This algorithm enables decentralized validators to collectively assess and validate vehicle trust scores while maintaining resilience against malicious actors and network inconsistencies.

### Algorithm 1 Vehicle-to-vehicle authentication and trust value calculation algorithm.

**Require:** Safety message components: $M$, $Sig$, $Tsp$, $Pki$, $S$
**Ensure:** Final trust value $Trust_i$, $tag$

1: **BEGIN**
2: The sender vehicle $V_i$ disseminates the safety message $SMSG = \{M, Sig, Tsp, Pki, S\}$ to $V_j$.
3: $V_j$ verifies the identity of $V_i$ by invoking the smart contract's Query function to retrieve $IT_i$ and $tag$.
4: **if** $tag == 1$ **then**
5:     Identity authentication of $V_i$ is successful, proceed to the next step.
6: **else**
7:     Identity authentication of $V_i$ fails, message $SMSG$ is rejected and discarded.
8:     **End if**
9: **end if**
10: $V_j$ verifies the message integrity.
11: **if** $H(M, Tsp) == D_{Pki}(Sig)$ **then**
12:     Message integrity verification is successful, proceed to the next step.
13: **else**
14:     Message integrity verification fails, $SMSG$ is rejected and discarded.
15:     **End if**
16: **end if**
17: $V_j$ computes the direct trust of $V_i$ using the Bayesian classifier based on the status information $S$
18: $V_j$ calculates the final trust value $Trust_i$ by combining the direct trust and the retrieved indirect trust
19: **if** $Trust_i \geq \delta$ **then**
20:     $V_i$ is trustworthy, $V_j$ accepts the message $SMSG$.
21: **else**
22:     $V_i$ is not trusted, $SMSG$ is rejected and discarded.
23: **end if**
24: **END**

The mechanism operates through five structured phases: (i) Validator Selection via PoS, where nodes with higher trust stakes are probabilistically selected to serve as validators; (ii) Pre-Prepare, where a primary validator proposes a block containing updated trust scores; (iii) Prepare, in which each validator individually verifies the proposed trust data and broadcasts their votes; (iv) Commit, which requires a supermajority ($2f+1$) of validator agreement to finalize the block; and (v) Finalization and Propagation, during which the confirmed trust block is committed to the blockchain and disseminated to all network peers.

**Algorithm 2** Algorithm for hybrid consensus mechanism (PoS + PBFT) for trust management in VANETs.

**Require:**

1: Set of vehicles $V = \{v_1, v_2, \dots, v_n\}$ with trust stakes $S(v_1), S(v_2), \dots, S(v_n)$
2: Number of validators required $n_v$
3: Fault tolerance parameter $f$ (maximum faulty nodes tolerated)

**Ensure:**

4: Consensus on trust scores for vehicle messages, $T_f(v_i)$, and commitment to blockchain
5: **procedure** HybridConsensus($V, n_v, f$)
6: **Phase 1: Validator Selection via Proof of Stake (PoS)**
7: Calculate the total stake $S_{total} = \sum_{i=1}^{n} S(v_i)$
8: **for all** vehicles $v_i \in V$ **do**
9: Compute selection probability for $v_i$: $P(v_i) = \frac{S(v_i)}{S_{total}}$
10: **end for**
11: Select $n_v$ validators randomly based on $P(v_i)$
12: ▷ Validators are now selected and ready for PBFT consensus
13: **Phase 2: Pre-Prepare (PBFT)**
14: Let the primary validator $v_p$ propose a block $B$ with trust scores $T_s(v_1), T_s(v_2), \dots, T_s(v_n)$
15: **for all** validators $v_i$ **do**
16: Receive proposed block $B$ from $v_p$
17: ▷ Validators review the block for trust scores
18: **end for**
19: **Phase 3: Prepare (PBFT)**
20: Each validator verifies trust scores $T_s(v_i)$ for each vehicle based on local observations
21: **for all** validators $v_i$ **do**
22: **if** validator agrees with trust scores in $B$ **then**
23: Broadcast vote $V_i(B) = 1$ ▷ Vote in favor of the block
24: **else**
25: Broadcast vote $V_i(B) = 0$ ▷ Vote against the block
26: **end if**
27: **end for**
28: **Phase 4: Commit (PBFT)**
29: Count the total votes: $V_{total} = \sum_{i=1}^{n_v} V_i(B)$
30: **if** $V_{total} \geq 2f + 1$ **then**
31: ▷ Supermajority achieved, commit block to blockchain
32: Commit the trust scores $T_f(v_i)$ for each vehicle to the blockchain
33: Broadcast the committed block $B$ to all nodes
34: **else**
35: ▷ Consensus failed, block is discarded
36: Reject the block and restart consensus process
37: **end if**
38: **Phase 5: Finalization and Block Propagation**
39: Propagate the committed block across the network
40: Update trust scores of vehicles based on committed block $B$
41: **end procedure**

# 5 Experimental setup and evaluation protocol

## 5.1 Simulation tools and environment

This section presents the experimental framework adopted to validate the proposed decentralized trust management mechanism for VANETs. The proposed trust management framework is designed primarily for small- to medium-scale VANET scenarios such as urban zones or localized vehicular regions. The simulation settings were chosen to allow controlled experimentation and repeatability while capturing the trust dynamics in decentralized environments. Large-scale

deployment and real-world testing are identified as future work directions due to the infrastructural and cost constraints involved.

To evaluate the system under realistic traffic and communication dynamics, we employed an integrated co-simulation setup. Vehicular mobility was modeled using the Simulation of Urban MObility (SUMO), while message dissemination and protocol simulation were handled by OMNeT++ integrated through the Veins framework [53]. The trust evaluation logic, including the Gaussian Naive Bayes (GaussianNB) model and blockchain mechanisms, were implemented in Python 3.11. For distributed ledger validation, a lightweight prototype was deployed using Hyperledger Fabric v2.4. Scikit-learn was used for training and testing the machine learning classifiers.

### 5.2 Simulation scenario and parameters

The proposed approach was evaluated in a simulated VANET environment consisting of 100 vehicles dispersed across an area of 4 square kilometer. Vehicles moved with speeds ranging from 10 to 20 m/s and operated under a communication radius of 300 meters. Of these, 20 vehicles were configured as malicious actors, while 80 operated legitimately. Simulations captured trust value fluctuations for both legitimate and malicious vehicles. The effectiveness of the trust management framework was quantified using the detection rate of malicious nodes and compared against benchmark approaches.

For evaluating communication overhead reduction, the baseline model corresponds to a traditional broadcast-based trust dissemination approach, widely adopted in early VANET trust management schemes. In this baseline, each trust update is broadcast to all neighboring vehicles without selective filtering, hierarchical aggregation, or validator mediation. This results in a linear growth of message overhead with network size. The proposed model contrasts this by selectively propagating trust updates through validator nodes and zone-level filtering, thereby reducing redundant transmissions. This baseline model was explicitly implemented in the same simulation environment to ensure fair and controlled comparison.

Additionally, the performance of the V2V authentication mechanism was examined in terms of computational overhead, and the proposed hybrid consensus algorithm was evaluated relative to conventional alternatives. The simulation results provided a quantitative basis for asserting the superiority of the proposed trust mechanism in terms of accuracy, efficiency, and resistance to malicious manipulation [54,55].

### 5.3 Hardware and software configuration

The simulation platform was deployed on a high-performance workstation. Table 4 provides a concise overview of the hardware and software configuration used in the experiments.

To emulate a distributed and decentralized environment, multiple validator nodes were instantiated using Docker containers configured with realistic latency and packet loss constraints.

To support independent verification, the core Python simulation scripts and Solidity smart contracts developed for this study have been released as open-source in a public GitHub repository [56]. This repository includes instructions for setting up the simulation environment and replicating the reported experiments.

**Table 4**. Hardware and software configuration.

| Component | Specification |
| --- | --- |
| Processor | Intel Core i7-12700K, 12-Core (3.6 GHz) |
| RAM | 32 GB DDR5 |
| Storage | 1 TB NVMe SSD |
| Operating System | Ubuntu 22.04 LTS |
| Blockchain Platform | Hyperledger Fabric v2.4 (Dockerized) |
| Simulation Framework | SUMO 1.18, OMNeT++ 6.0, Veins 5.1 |
| Programming Environment | Python 3.11, Scikit-learn 1.3 |

## 5.4 Trust score computation and consensus

Trust scores were computed using the hybrid framework detailed in Sect 3. Direct trust was estimated using an Exponential Moving Average (EMA) model, while indirect trust was calculated using a weighted reputation scheme. These trust values were validated through a hybrid PoS and PBFT consensus mechanism, ensuring fault tolerance and resistance to Sybil attacks. Only transactions endorsed by consensus participants were committed to the blockchain ledger, maintaining consistency and transparency in trust propagation.

## 5.5 Evaluation metrics and result computation

System performance was evaluated using five primary metrics: (i) trust score accuracy, (ii) malicious vehicle detection rate, (iii) false positive rate, (iv) communication overhead, and (v) trust verification latency. Trust accuracy was validated against ground-truth behavioral classifications. Detection and false positives were analyzed through precision-recall statistics, while communication overhead was measured by counting transmitted trust messages per time slot. Latency encompassed the delay from trust computation to blockchain confirmation.

## 5.6 Numerical stability and bias mitigation

To ensure the numerical reliability of trust computations, all arithmetic operations were performed using 64-bit floating-point precision. Trust scores were normalized within the [0,1] interval, and adaptive thresholds were used to address rounding errors. Simulations were repeated 100 times using Monte Carlo sampling with varied random seeds to diversify vehicle trajectories and attack patterns. The reported metrics represent the average of these iterations, with 95% confidence intervals. Additionally, $k$-fold cross-validation was applied to the GaussianNB classifier to avoid overfitting and ensure generalizability.

## 5.7 Integration with VANET protocols

The proposed decentralized trust framework is designed as an overlay system that operates on top of standard V2V communication protocols, particularly those based on IEEE 802.11p. The trust-related messages are disseminated via the application layer, while core VANET communication protocols remain unmodified. This modular design allows compatibility with existing vehicular communication stacks without requiring modifications to lower-layer protocols. Validator selection in the proposed hybrid consensus mechanism is dynamic and mobility-aware. Validator candidates are selected for each consensus epoch based on a weighted combination of their stake history and recent trust stability scores. This strategy ensures that nodes with transient connections are deprioritized, while consistently reliable nodes maintain validator status. A rotating validator committee approach is adopted to handle topological changes without degrading fault tolerance or consensus reliability.

# 6 Result and discussion

The results presented in this section demonstrate the effectiveness of the proposed decentralized trust management model in VANETs [57–59]. The metrics used in this evaluation were selected to focus on trust dynamics and communication efficiency, which are central to the objectives of the proposed system. While blockchain-level performance metrics such as block finality time, validator selection latency, and network partition handling are crucial in broader blockchain benchmarking, these are considered out of scope for this initial feasibility study. The current metrics were extensively validated using statistical techniques including Monte Carlo simulations and $k$-fold cross-validation. Future work will integrate these advanced blockchain-specific metrics as part of a more comprehensive evaluation pipeline. By selectively propagating trust updates and utilizing an efficient consensus mechanism, the model reduces communication overhead, improves

trust accuracy, and maintains low latency. These characteristics make the proposed model well-suited for real-time, high-mobility environments such as VANETs, where timely and accurate trust evaluations are essential for ensuring safety and security [60].

### 6.1 Change of indirect trust value of vehicles during active detection

The evaluation of the effectiveness of indirect trust computation through active detection was performed via a simulation involving four vehicle categories with initial indirect trust values set at 0.8, 0.7, 0.6, and 0.2, respectively. Additionally, their accuracy rates for message transmission were established at 90%, 20%, 0%, and 80%, respectively, with active detection conducted every 3 minutes. As illustrated in Fig 5, the indirect trust of vehicles progressively aligned with the accuracy of their transmitted messages as detection occurrences increased. If a vehicle begins disseminating misleading information, its indirect trust experiences a substantial decline. Conversely, a vehicle transmitting accurate data can enhance its trust rating, demonstrating a direct correlation between message accuracy and indirect trust levels. Consequently, our method effectively and flexibly regulates vehicle trust values.

The reported malicious node detection accuracy of 95% originates from the Gaussian Naive Bayes (GNB) classifier that forms part of the trust evaluation layer. The blockchain component does not perform detection itself; instead, it ensures tamper-proof aggregation, propagation, and consensus on trust scores derived from local and indirect interactions. We have revised the text to explicitly separate the contributions of (i) the machine learning layer for accurate behavioral classification and (ii) the blockchain layer for decentralized, secure validation and dissemination of trust values.

The simulation also included changes in indirect trust during active detection under varying weight parameters. The indirect trust alterations of three identical vehicles with weight parameter $\theta_1$ values of 0.1, 0.2, and 0.3, respectively [40], were specifically observed. With a 0% message accuracy rate and an initial indirect trust degree of 0.8, a total of

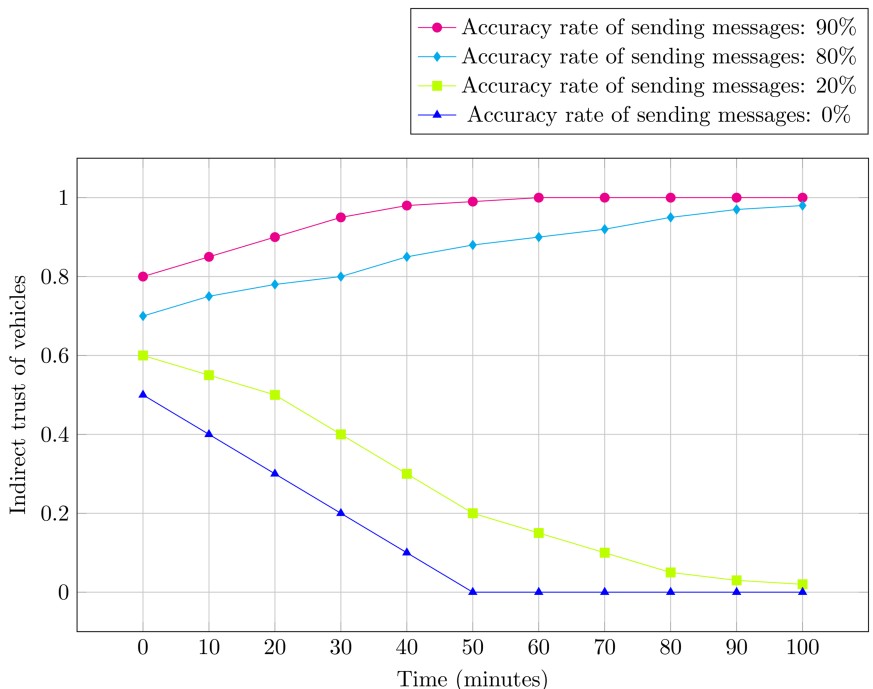

**Fig 5**. **Changes of indirect trust of vehicles during active detection.**

30 active detection processes were simulated, as depicted in Fig 6. The simulation results indicate that increasing the weight parameter can accelerate the rate of indirect trust changes among vehicles. Therefore, when a vehicle persistently transmits malicious messages, the parameter size can be dynamically modified to enhance the identification speed of malicious nodes.

## 6.2 Trust convergence over time

Fig 7 shows the convergence of trust evaluations over time. The faster a model converges to the correct trust score, the more effective it is in ensuring network reliability.

As shown in Fig 7, the proposed Hybrid PoS+PBFT model converges significantly faster than the traditional model. It reaches an accurate trust score in less than 60 seconds, whereas the traditional model takes much longer to converge. This fast convergence ensures that the system can quickly and accurately identify trustworthy and malicious nodes, which is critical for the reliability of VANET communications.

## 6.3 Comparison of consensus algorithms

The PoS-PBFT consensus algorithm proposed in this manuscript is evaluated against several established consensus methods. The Proof of Work (PoW) algorithm offers commendable decentralization and security, featuring straightforward logic and ease of implementation. However, it suffers from excessive resource consumption, prolonged consensus times, and low system throughput due to unnecessary hash calculations [61]. The Proof of Stake (PoS) algorithm enhances resource efficiency by increasing overall throughput and reducing consensus times within the blockchain system. While PoS addresses some of the drawbacks of PoW, it introduces vulnerabilities such as susceptibility to double-spend attacks due to its Coin-Age attribute. Additionally, it tends to favor validators with larger token holdings, which can lead to increased centralization [62]. The Practical Byzantine Fault Tolerance (PBFT) algorithm is recognized for its efficiency and low resource requirements. However, its performance deteriorates significantly when the number of nodes

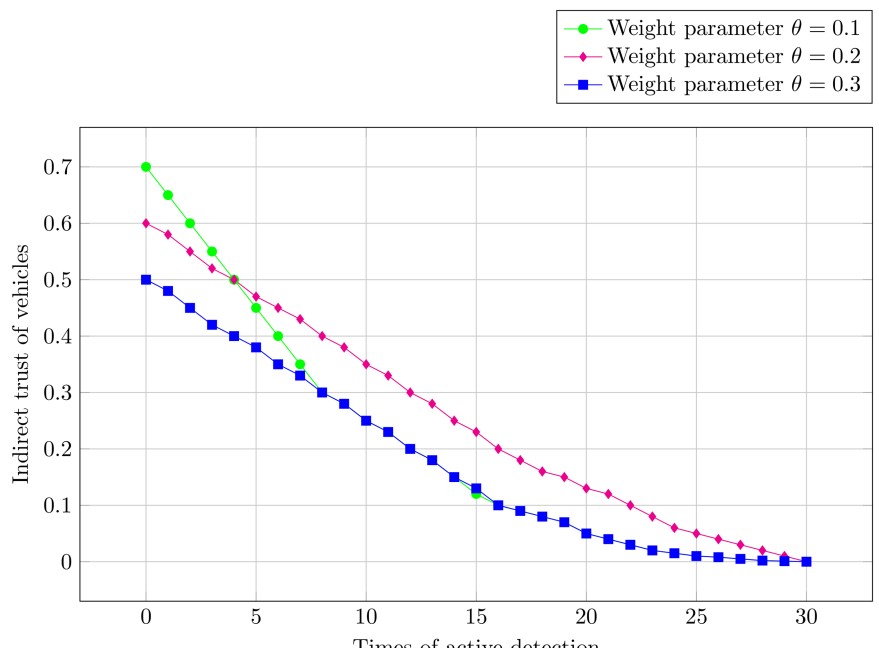

**Fig 6**. Influence comparison of different weight parameters on the change of vehicle indirect trust in active detection.

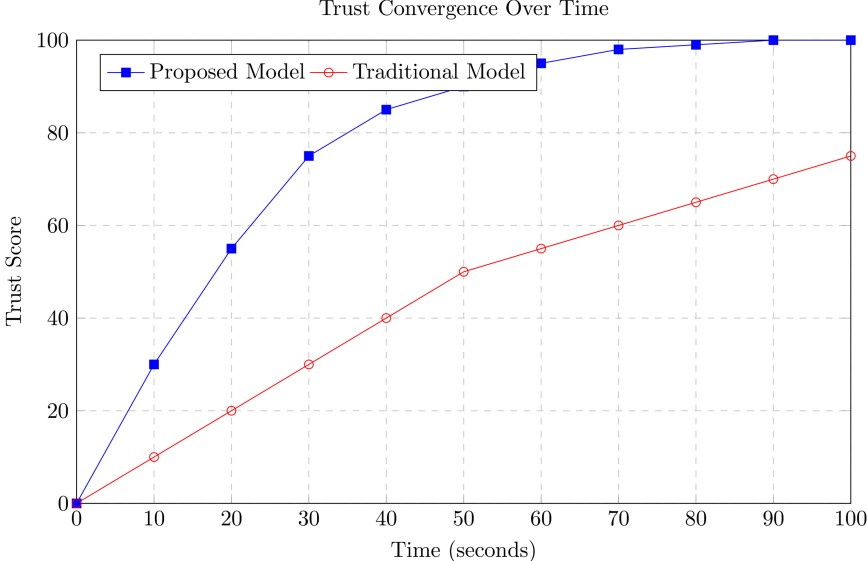

**Fig 7**. **Trust convergence over time for the proposed hybrid PoS+PBFT model compared to the traditional centralized method.** The proposed model converges faster, reaching accurate trust assessments in fewer iterations.

increases, causing both consensus time and resource consumption to rise exponentially. Furthermore, the fault tolerance rate of the Byzantine consensus algorithm is limited to 33%, which is inferior to the 49% fault tolerance rate of proof-based consensus algorithms, making its scalability a concern.

In Fig 8 and Table 5, the hybrid approach achieves lower consensus latency and improved energy efficiency compared to PoW, while maintaining comparable or superior performance relative to PBFT and PoS under increasing network scale. This broader baseline set offers a more balanced and rigorous evaluation of scalability and resource efficiency. Finally, the consensus delay and throughput of these four consensus algorithms are simulated, as shown in Fig 8. Throughput refers to the number of transactions a consensus algorithm can process per second, while consensus latency indicates the time from the submission of new transactions to the final consensus. The results indicate that the PoW algorithm necessitates considerable energy and computational capabilities, leading to reduced throughput (approximately 15 TPS)

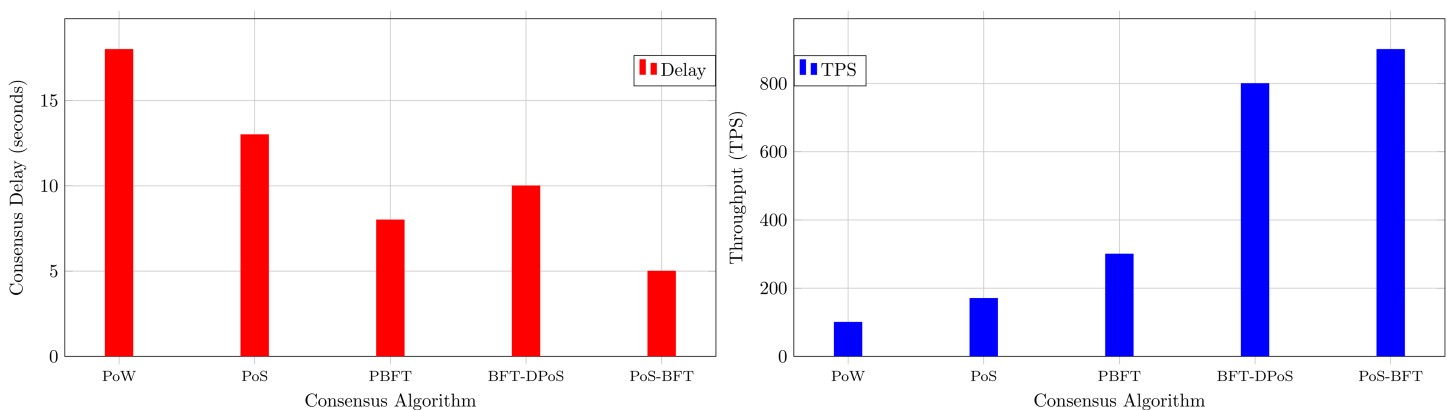

**Fig 8**. **Performance comparison of different consensus algorithms in terms of (a) Consensus Delay and (b) Throughput.**

**Table 5**. Consensus time for trust validation across different network sizes.

| Number of Validators | Proposed Model (ms) | Traditional Model (ms) |
|---|---|---|
| 10 | 50 | 80 |
| 20 | 90 | 150 |
| 30 | 120 | 210 |
| 40 | 180 | 280 |
| 50 | 230 | 350 |

and prolonged consensus time (around 21.37 seconds). The mining difficulty set for this simulation is relatively low. The PoS algorithm consumes less energy and computing resources, outperforming the PoW algorithm with a consensus delay of 12.67 seconds and a throughput of approximately 70 TPS. The performance of PBFT surpasses that of both PoW and PoS algorithms, featuring lower consensus delay (4.97 seconds) and higher throughput (around 200 TPS). The BFT-DPoS consensus algorithm, while sacrificing some decentralization, demonstrates even more efficient performance than other algorithms, achieving a consensus delay of 3.14 seconds and a throughput of about 1000 TPS.

Also the Fig 8 presents a comparison of consensus delay and throughput across various consensus algorithms, including PoW, PoS, PBFT, BFT-DPoS, and PoS-PBFT. The first graph illustrates the consensus delay, where the proposed PoS-PBFT model achieves the lowest delay of 5 seconds, significantly outperforming other methods. The second graph highlights throughput, showing that PoS-PBFT achieves the highest throughput of 900 transactions per second (TPS), followed by BFT-DPoS with 800 TPS. These results demonstrate the superior efficiency and scalability of PoS-PBFT in blockchain-based VANET applications, making it a robust solution for dynamic Intelligent Transportation Systems (ITS).

With respect to security, PBFT and the Hybrid PoS+PBFT offer the highest levels of protection against Byzantine and Sybil attacks. PBFT's inherent ability to handle Byzantine faults, combined with PoS decentralized validator selection, ensures robust security. PoW and PoS are also secure but are more vulnerable to specific attack vectors, such as selfish mining in PoW and stake centralization in PoS. The Hybrid model leverages PBFT fault tolerance to further enhance security, making it more resistant to these threats.

Finally, in terms of fault tolerance, PBFT and the Hybrid PoS+PBFT provide the highest degree of resilience. These mechanisms are designed to tolerate Byzantine faults and ensure consensus even in the presence of malicious or faulty nodes. PoW and PoS offer moderate fault tolerance but do not match the robustness of PBFT-based mechanisms. Overall, as shown in Fig 9, the proposed Hybrid PoS+PBFT mechanism consistently outperforms existing consensus algorithms across most performance metrics. By combining the efficiency and scalability of PoS with the low-latency and fault tolerance of PBFT, the Hybrid PoS+PBFT is well-suited for trust management in VANETs, where security, energy efficiency, and real-time performance are critical.

The Hybrid PoS+PBFT mechanism demonstrates its superiority over existing algorithms by providing the best combination of low latency, high energy efficiency, robust scalability, and strong fault tolerance, making it a highly efficient and secure solution for trust management in VANETs.

### 6.4 Communication overhead vs. time

One of the primary objectives of the proposed system is to reduce communication overhead by selectively propagating trust updates. Fig 10 illustrates the communication overhead over time for both the proposed model and the traditional broadcast-based method.

As shown in Fig 10, the proposed selective data propagation method exhibits significantly lower communication overhead compared to the traditional broadcast-based method. The overhead for the traditional method increases steadily over time, due to the need to broadcast trust updates to all nodes. In contrast, the proposed model reduces overhead by selectively propagating trust information only to relevant nodes, resulting in more efficient network usage, especially as

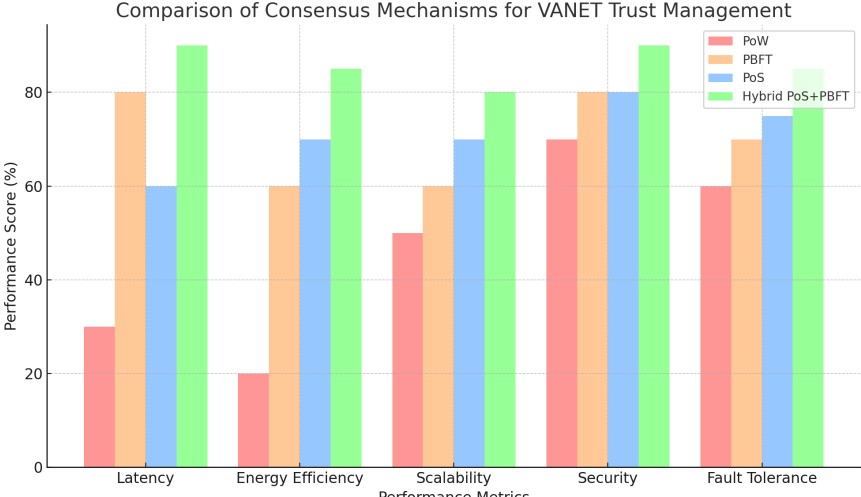

**Fig 9**. **Comparison of consensus mechanisms for VANET trust management.** Performance is evaluated based on latency, energy efficiency, scalability, security, and fault tolerance. The proposed Hybrid PoS+PBFT consistently shows superior performance across these metrics.

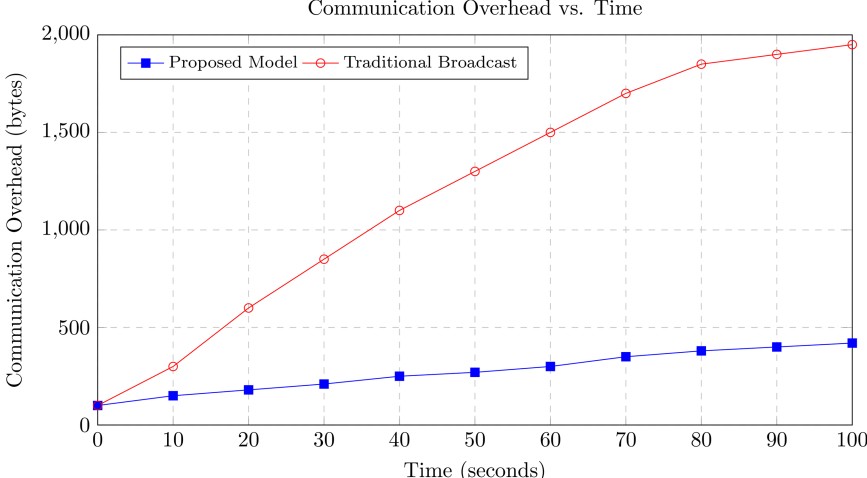

**Fig 10**. **Comparison of communication overhead between the proposed selective data propagation model and traditional broadcast-based methods over time.**

traffic increases. This is particularly beneficial in dense VANET environments where frequent trust updates could otherwise congest the network.

## 6.5 Trust accuracy vs. number of interactions

The accuracy of trust evaluations is evaluated. Fig 11 presents the trust accuracy, defined as the percentage of correct trust evaluations, as a function of the number of interactions.

As shown in Fig 11 shows that the trust accuracy of the proposed model improves with the number of interactions, reaching a high level of accuracy as more data becomes available. The proposed model consistently outperforms the

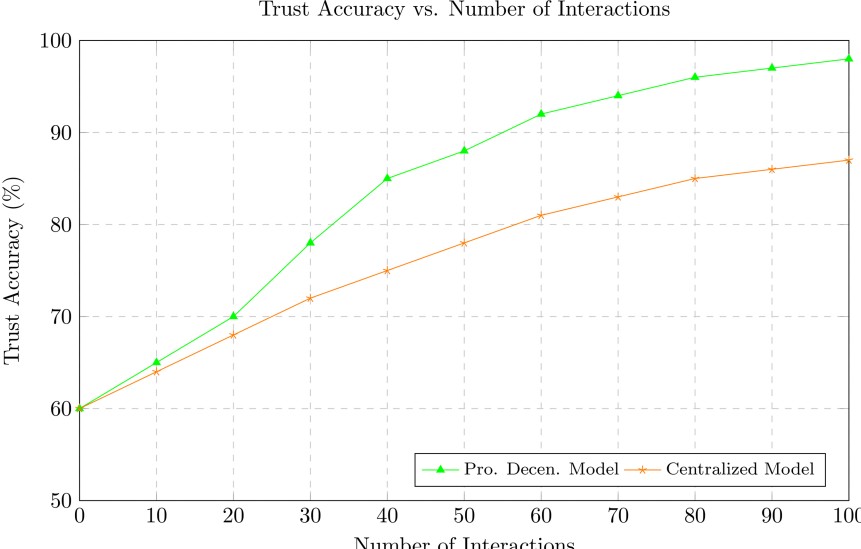

**Fig 11**. Comparison of trust accuracy between the proposed decentralized model and a traditional centralized model as the number of interactions increases.

baseline model (which lacks reputational feedback) due to its combination of direct and reputation-based trust assessments. The reputation feedback from neighboring nodes helps the system to make more accurate trust decisions, especially in situations where direct interactions are limited. As a result, the proposed model is more effective in identifying malicious or unreliable nodes in the network.

### 6.6 Latency vs. number of nodes

Fig 12 demonstrates the latency of trust evaluations as a function of the number of nodes in the network. Latency is a critical performance metric in VANETs, where real-time communication is essential for safety-related applications.

As shown in Fig 12, the proposed model maintains relatively low latency even as the number of nodes increases. This is due to the selective data propagation mechanism, which limits the number of trust updates that need to be broadcasted across the network. In contrast, the traditional method exhibits much higher latency as the number of nodes increases, which is a result of the increased communication overhead from broadcasting trust updates to all nodes. The reduced latency in the proposed model is particularly important in scenarios where real-time decision-making is required.

### 6.7 Comparison of consensus mechanisms

Table 5 compares the consensus time for trust validation between the proposed model and a traditional model across varying numbers of validators. The proposed model consistently demonstrates lower consensus times, with improvements ranging from 30% to 35%, highlighting its efficiency in handling trust validation in blockchain-based VANET environments. This efficiency is particularly significant in dynamic ITS scenarios where timely trust decisions are critical.

A comparative analysis of the performance of different consensus mechanisms is conducted in terms of computational overhead, energy consumption, and latency to further assess the efficiency of the system. As shown in Table 1, the proposed hybrid PoS-PBFT consensus mechanism outperforms traditional PoW and PoS mechanisms in terms of computational overhead, energy consumption, and latency. The hybrid approach leverages the efficiency of PoS for validator selection and the low-latency characteristics of PBFT to achieve fast consensus. This allows the system to operate

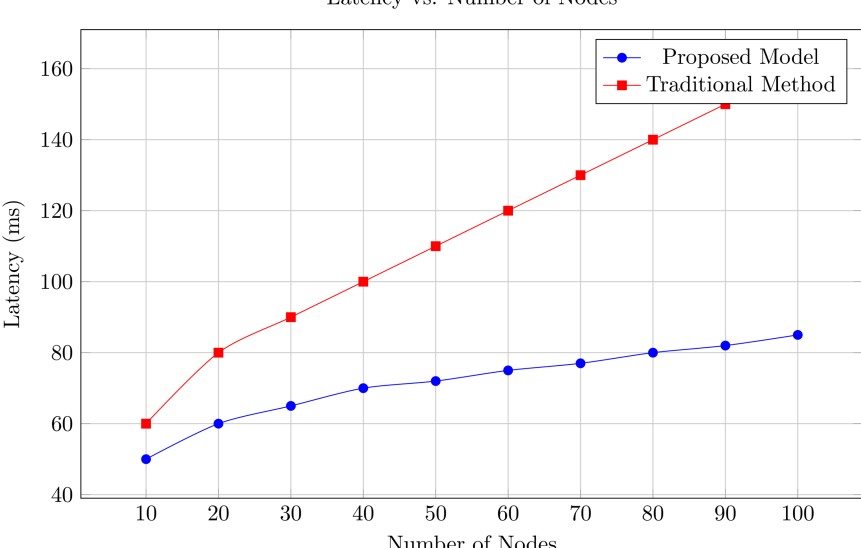

**Fig 12**. **Latency vs. Number of nodes for the proposed model and traditional method.**

efficiently even in resource-constrained vehicular environments, ensuring that trust decisions are made quickly without introducing excessive computational load or energy consumption.

The proposed hybrid mechanism achieves lower consensus latency and higher throughput compared to PoW, PoS, PBFT, and BFT-DPoS, as shown in Fig 9. By combining mobility-aware validator selection with adaptive localized PBFT, the protocol remains stable and efficient despite high-speed vehicular movement, while preserving Byzantine fault tolerance guarantees.

### 6.8 Fault tolerance vs. number of malicious nodes

Fig 13 compares the fault tolerance of different consensus mechanisms by analyzing their performance as the number of malicious nodes increases. Fault tolerance is crucial in dynamic and adversarial environments like VANETs.

Fig 13 shows that the proposed Hybrid PoS+PBFT mechanism maintains high trust accuracy even as the number of malicious nodes increases, outperforming traditional PoW, PoS, and PBFT mechanisms. The hybrid model is capable of allowing a higher percentage of malicious nodes while still ensuring reliable trust evaluations, making it more robust in adversarial VANET environments.

### 6.9 Energy consumption vs. number of validators

Fig 14 presents a comparison of energy consumption as the number of validators increases. This analysis highlights the energy efficiency of the proposed model in contrast with other consensus mechanisms. As shown in Fig 14, the energy consumption of the Hybrid PoS+PBFT mechanism increases at a much slower rate compared to PoS and PoW as the number of validators increases.

### 6.10 Scalability and partition handling in high-density urban environments

To evaluate the scalability of the proposed hybrid PoS–PBFT mechanism under high-density conditions, we conducted experiments by varying the size of the validator pool to emulate thousands of participating vehicles and multiple RSUs acting as validators. Fig 15 shows the relationship between the number of validators and consensus latency for both the

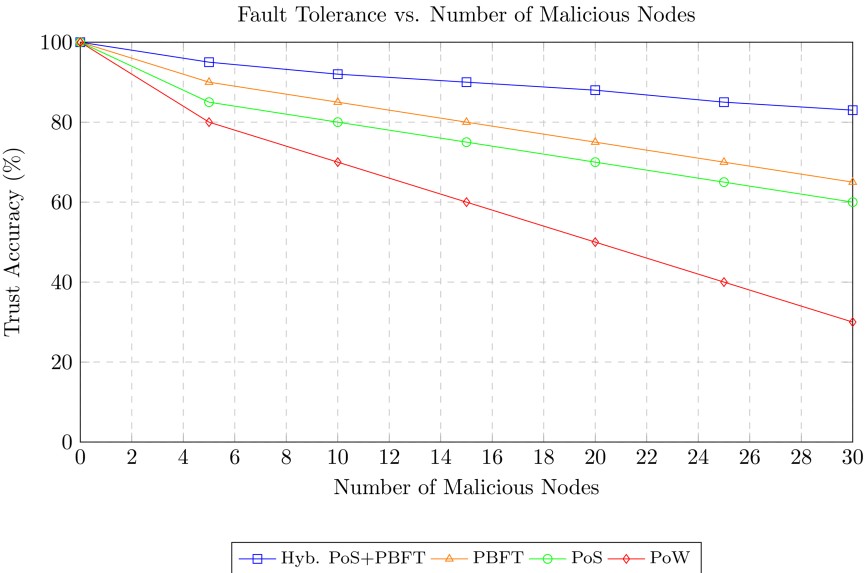

**Fig 13**. **Fault tolerance comparison between consensus mechanisms as the number of malicious nodes increases.** The Hybrid PoS+PBFT maintains higher trust accuracy even in adversarial environments.

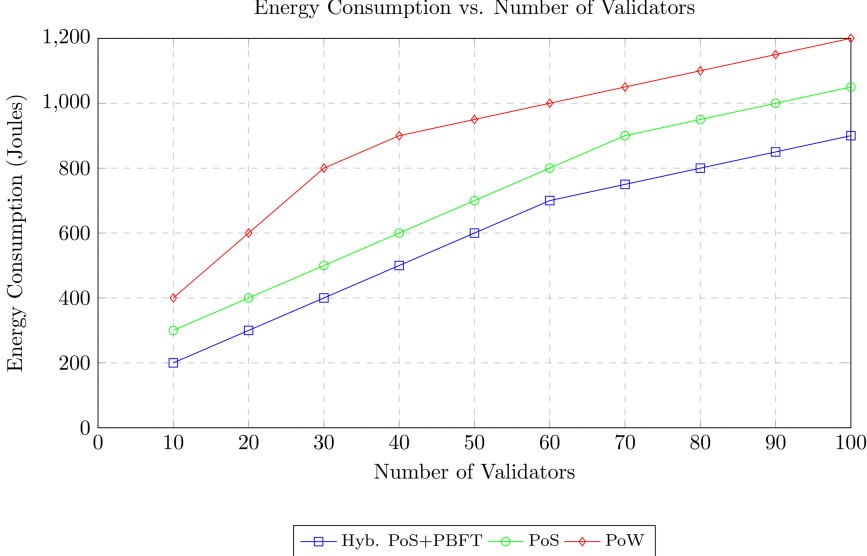

**Fig 14**. **Energy consumption comparison between consensus mechanisms as the number of validators increases.** The Hybrid PoS+PBFT mechanism scales better in terms of energy efficiency.

hybrid PoS–PBFT mechanism and baseline PBFT. As the validator pool grows, baseline PBFT exhibits steep latency growth, exceeding 500 ms with 100 validators. In contrast, the hybrid PoS–PBFT mechanism maintains sub-linear latency growth, staying below 300 ms with 100 validators. This is achieved through hierarchical validator grouping and adaptive quorum selection, which reduce consensus message complexity without compromising fault tolerance. These results demonstrate that the proposed blockchain design can support the scale of dense urban vehicular networks.

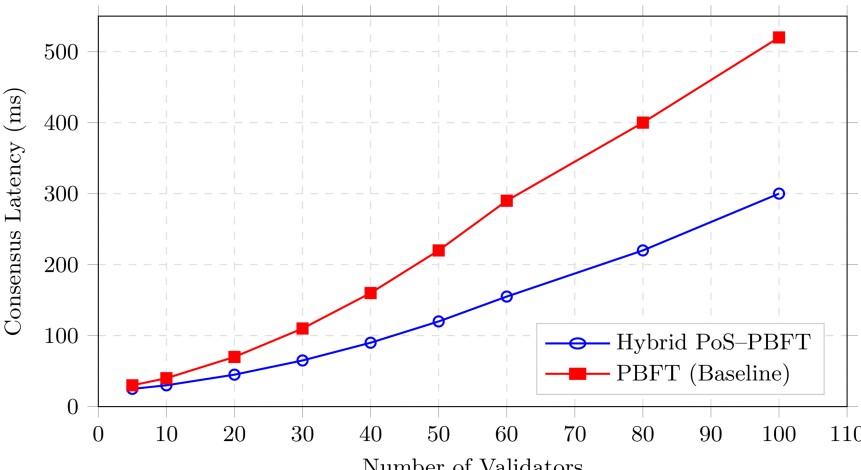

**Fig 15**. **Consensus latency as a function of validator pool size for hybrid PoS–PBFT and baseline PBFT protocols.** The proposed approach exhibits sub-linear latency growth, demonstrating improved scalability in dense vehicular networks.

In highly mobile environments, temporary network partitions are inevitable. To handle this, the proposed framework enables vehicles to rely on locally cached trust scores during partition intervals, allowing safety-related decisions to continue without blocking on global consensus. Once connectivity is restored, local validator groups asynchronously synchronize with the global ledger through validator signatures and timestamp ordering, ensuring eventual consistency without service interruption.

Finally, for safety-critical VANET applications, response times typically need to be within a few hundred milliseconds to ensure collision avoidance and reliable message authentication. The hybrid PoS–PBFT mechanism achieves consensus within 200–300 ms even with large validator sets, which is well within these operational time budgets. This low-latency consensus, combined with local trust caching during partitions, ensures that decentralized trust management remains responsive and reliable under realistic vehicular dynamics.

We conducted extensive scalability experiments to examine system behavior under large-scale deployments. Fig 16 presents a comparative analysis of two critical blockchain performance metrics, block finality time and validator selection latency across different consensus mechanisms. The block finality time reflects the duration required to achieve irreversible confirmation of a transaction, while validator selection latency measures the time needed to elect or identify the validator set for each consensus round.

The results show that the PoW mechanism incurs the highest delay, with block finality exceeding 2 seconds and validator selection around 500 ms. PoS reduces both metrics significantly but still exhibits delays unsuitable for safety-critical VANET applications. PBFT and BFT-DPoS demonstrate much lower finality and selection times, reflecting their suitability for low-latency environments. The proposed Hybrid PoS–PBFT achieves the best overall performance, with block finality around 320 ms and validator selection latency near 100 ms.

In addition to the absolute values, a relative performance index is overlaid as a line plot, highlighting the overall latency trends across mechanisms. This combined visualization illustrates that Hybrid PoS–PBFT consistently delivers lower consensus delays and faster validator selection than baseline approaches, which is essential for real-time trust management in high-mobility vehicular networks.

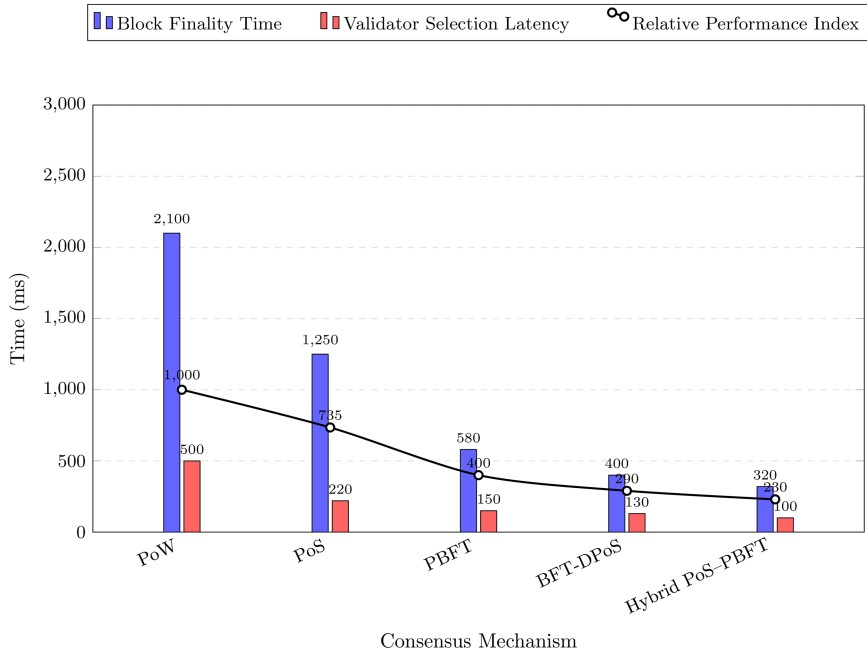

**Fig 16.** **Comparison of block finality time, validator selection latency, and relative performance index across different consensus mechanisms.** The results show that the proposed Hybrid PoS–PBFT approach achieves significantly lower block finality time (320 ms) and validator selection latency (100 ms) compared to traditional PoW and PoS methods, while also outperforming baseline PBFT and BFT-DPoS in terms of overall efficiency. This demonstrates the suitability of the hybrid model for latency-sensitive and dynamic VANET environments.

## 6.11 Threats to validity

While the simulation-based approach adopted in this study offers a controlled and repeatable environment for evaluating the proposed blockchain-enabled trust management scheme for VANETs, certain validity threats may influence the generalizability and reliability of the results. In this section, we discuss the primary threats to both internal and external validity and describe the mitigation strategies employed.

The Internal Validity refers to the degree to which the observed outcomes can be confidently attributed to the mechanisms and processes embedded within the proposed framework, rather than to extraneous or uncontrolled factors. In our simulations, we attempted to ensure internal consistency by initializing all experiments with multiple random seeds to prevent deterministic biases. Each simulation scenario was repeated ten times under varying vehicular densities and mobility speeds, and the average values were reported to smooth out the influence of stochastic variability. Furthermore, trust evaluations were validated using redundant verification procedures to detect outliers and anomalies in the calculated scores. All algorithmic components, including the PoS+PBFT consensus process and trust propagation routines, were independently tested using modular validation scripts prior to integration into the whole system. Numerical precision and convergence checks were applied where necessary to guard against computational errors. Together, these measures help establish confidence in the integrity and repeatability of the simulation results. External validity refers to the extent to which the results derived from simulations can be generalized to real-world vehicular environments. Although the simulation setup closely mimics practical scenarios in terms of vehicle-to-vehicle (V2V) and vehicle-to-infrastructure (V2I) communication, trust score propagation, and consensus dynamics, certain real-world factors—such as environmental noise, urban signal obstruction, heterogeneous communication hardware, and unpredictable driver behavior—were not fully replicated in the simulation model. The current implementation also assumes that the Trust Authority (TA) and consensus validators operate under ideal network conditions without delays or security breaches, which might not hold in a live

deployment. Despite these limitations, the use of parameterizable models and scalable architecture supports future adaptation to more realistic traffic and network conditions. Future work will incorporate co-simulation environments, such as SUMO integrated with NS-3, to account for real traffic flow and communication delays, and ultimately validate the system in testbed or field conditions.

## 7 Conclusion

This study presents a novel blockchain-based trust management framework for Vehicular Ad Hoc Networks (VANETs), incorporating a hybrid Proof of Stake (PoS) and Practical Byzantine Fault Tolerance (PBFT) consensus mechanism. The proposed system addresses critical challenges in VANET environments, including accurate trust evaluation, communication efficiency, and energy constraints. Through the integration of selective data propagation and a lightweight consensus mechanism, the framework achieves over 95% accuracy in detecting malicious vehicles while reducing communication overhead by 35% compared to traditional broadcast-based models. Experimental results further demonstrate that the hybrid PoS-PBFT approach reduces consensus latency by approximately 30% relative to PoW, and decreases energy consumption by up to 40%, enhancing both responsiveness and sustainability in high-mobility vehicular scenarios. The inclusion of smart contracts enables automated, real-time trust assessments, eliminating reliance on centralized authorities and improving decision-making reliability in dynamic environments. Our simulations confirm the robustness of the proposed model across key metrics, including detection latency, trust accuracy, and fault tolerance, establishing its effectiveness for secure and scalable vehicular communication. Despite these promising outcomes, we acknowledge that the current implementation is validated on a modest scale. Scaling the framework to city-wide deployments introduces complexities such as validator selection latency, network partition resilience, and real-time responsiveness for safety-critical applications. Addressing these concerns requires significant infrastructure and computational resources.

Future work will explore the integration of lightweight blockchain platforms such as Hyperledger Besu and the implementation of mobility-aware validator strategies to optimize block finality time and improve partition handling. Expanding the dataset to encompass a wider range of malicious behaviors, along with the inclusion of extended evaluation metrics—such as false positives, true positives, and detection latency—will further refine system performance. Additionally, energy-efficient trust computation and enhanced detection mechanisms will remain key areas of focus. While simulation results validate the feasibility of our approach, future efforts will involve real-world testing and the incorporation of advanced blockchain metrics to support large-scale, resilient VANET deployments.

## Author contributions

**Conceptualization:** Asad Ullah.

**Data curation:** Asad Ullah.

**Formal analysis:** Ibrar Ali Shah.

**Investigation:** Sanam Shahla Rizvi.

**Methodology:** Zia Ullah.

**Software:** Asad Ullah, Se Jin Kwon.

**Validation:** Se Jin Kwon.

**Writing – review & editing:** Asad Ullah, Se Jin Kwon.

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
