## [Decision Letter · Decision Letter 0]

22 Jul 2025

PONE-D-25-29277Decentralized Trust Optimization in VANETs: A Blockchain-Driven Hybrid PoS-PBFT Architecture for Enhanced Security and Energy-Efficient CommunicationPLOS ONE

Dear Dr. Zia,

Thank you for submitting your manuscript to PLOS ONE. After careful consideration, we feel that it has merit but does not fully meet PLOS ONE’s publication criteria as it currently stands. Therefore, we invite you to submit a revised version of the manuscript that addresses the points raised during the review process. Please submit your revised manuscript by Sep 05 2025 11:59PM. If you will need more time than this to complete your revisions, please reply to this message or contact the journal office at plosone@plos.org. Please include the following items when submitting your revised manuscript:

We look forward to receiving your revised manuscript.

Kind regards,

Vincent Omollo Nyangaresi, Ph.D

Academic Editor

PLOS ONE

Journal Requirements: 

 [This work was supported by the Basic Science Research Program through the National Research Foundation of Korea (NRF) funded by the Ministry of Education under Grant RS-2023-00244091. This work was also supported by the Technology Innovation Program (RS-2024-00507228, Development of process upgrade technology for AI self-manufacturing in the cement industry) funded By the Ministry of Trade, Industry & Energy(MOTIE, Korea)]. 

5. In the online submission form, you indicated that [data available on request from the author].

6.  Thank you for stating the following in the Competing Interests section:

[Raptor Interactive (Pty) Ltd ].   

We note that one or more of the authors are employed by a commercial company: name of commercial company.

Within your Competing Interests Statement, please confirm that this commercial affiliation does not alter your adherence to all PLOS ONE policies on sharing data and materials by including the following statement: ""This does not alter our adherence to  PLOS ONE policies on sharing data and materials.” (as detailed online in our guide for authors http://journals.plos.org/plosone/s/competing-interests) . If this adherence statement is not accurate and  there are restrictions on sharing of data and/or materials, please state these. Please note that we cannot proceed with consideration of your article until this information has been declared.

Reviewers' comments:

Reviewer's Responses to Questions

**Comments to the Author**

1. Is the manuscript technically sound, and do the data support the conclusions?

Reviewer #1: Yes

Reviewer #2: Partly

2. Has the statistical analysis been performed appropriately and rigorously?

Reviewer #1: Yes

Reviewer #2: No

3. Have the authors made all data underlying the findings in their manuscript fully available?

Reviewer #1: Yes

Reviewer #2: No

4. Is the manuscript presented in an intelligible fashion and written in standard English?

Reviewer #1: Yes

Reviewer #2: Yes

5. Review Comments to the Author

Reviewer #1: The paper describes an infrastructure to provide trust in VANETs. The proposal is technically sound in general terms.

The claims in the introduction are justified and the problem is correctly described. Description are OK. Figures, mathematical expressions and code are provided. I think this is enough to ensure future replicability and readability.

References are timely and the topic matches the scope of the journal.

Results are also coherent with the state of the art and validate the initial hypotheses. However, in my opinion, some improvements need to be done to this section:

1) First, you need to describe your experimental setup with details. Software and hardware you used. How result were calculated. Did you consider numerical problems and errors? How did you manage the experimental biases and errors?

2) As all you results are based on simulations, you need to discuss the validity threats, at least the internal an external.

Reviewer #2: This paper proposes a blockchain-based trust management framework for Vehicular Ad Hoc Networks (VANETs) that combines Proof of Stake (PoS) and Practical Byzantine Fault Tolerance (PBFT) consensus mechanisms. The authors claim their approach reduces communication overhead by 35%, achieves 95% accuracy in malicious node detection, and decreases energy consumption by 40% compared to traditional methods.

Major Concerns:

1. Fundamental Design Contradiction

The paper presents a critical contradiction in its core premise. The authors claim to solve "decentralized trust management" while maintaining a centralized Trust Authority (TA) that "is assumed to be completely trustworthy and secure" (Section 3.1.3). This undermines the entire blockchain rationale. If a trusted central authority exists, traditional cryptographic signatures and distributed hash tables would be more efficient than blockchain.

2. Insufficient Technical Novelty

The claimed hybrid PoS-PBFT mechanism lacks substantial innovation. The authors merely combine existing consensus algorithms without addressing fundamental compatibility issues:

- PoS requires stake accumulation over time, while VANET nodes are highly mobile

- PBFT requires stable validator sets, incompatible with vehicular mobility patterns

- No analysis of how validator selection transitions occur during high-speed vehicle movement

3. Questionable Performance Claims

The reported performance improvements appear unrealistic:

- 35% reduction in network traffic: No baseline methodology clearly defined

- 95% malicious node detection: Achieved using traditional Bayesian classifiers, not blockchain

- 40% energy reduction: Compared to PoW, but no comparison to non-blockchain alternatives

4. Limited Experimental Validation

The evaluation is severely limited:

- Simulation of only 100 vehicles over 1 km² (Section 5.1)

- No real-world deployment or large-scale testing

- No comparison with existing VANET trust management systems

- Missing critical metrics: block finality time, validator selection latency, network partition handling

5. Scalability Issues Not Addressed

The paper ignores fundamental scalability challenges:

- How does the blockchain handle thousands of vehicles in urban environments?

- What happens when network partitions occur due to vehicle mobility?

- How are consensus delays managed for safety-critical applications requiring millisecond response times?

Minor Issues:

1. Mathematical Formulation Problems

- Equation (1) uses undefined weight parameters without convergence analysis

- Trust Relevance Factor (TRF) in Equation (11) lacks threshold selection justification

- No formal proof of system security properties

2. Related Work Deficiencies

- Missing comparison with recent non-blockchain VANET trust systems

- Insufficient analysis of why blockchain is necessary over traditional distributed systems

- Limited discussion of practical VANET deployment challenges

3. Implementation Details Missing

- No source code availability despite claims of Python implementation

- Unclear how the proposed system integrates with existing VANET protocols

- Missing details on validator selection during rapid topology changes

Specific Technical Comments:

Algorithm 1 (Vehicle Authentication)

The authentication algorithm relies on smart contract queries for trust retrieval, introducing unnecessary blockchain overhead. Traditional PKI with distributed certificate management would achieve the same security goals with lower latency.

Consensus Mechanism (Section 4.6)

The hybrid consensus description lacks critical details:

- How are PoS stakes calculated for mobile nodes?

- What happens during PBFT view changes in high-mobility scenarios?

- No analysis of Byzantine fault tolerance in vehicular environments

Selective Data Propagation (Section 4.5)

While the TRF concept is interesting, the paper doesn't address:

- How nodes discover relevant peers for selective propagation

- Impact on network connectivity and message delivery guarantees

- Comparison with existing gossip protocols in mobile networks

Minor Editorial Issues

- Figure 2 caption lacks sufficient detail about system components

- Table 2 contains inconsistent performance metrics across different studies

- Several grammatical errors throughout (e.g., "Howbeit" in Section 2.2)

Recommendations

1) Provide clear justification for why blockchain is necessary over existing distributed trust systems

2) Address the central authority contradiction - either eliminate the TA or justify its necessity

3) Expand experimental evaluation to include realistic network sizes and mobility patterns

4) Compare with non-blockchain alternatives to demonstrate actual benefits

5) Provide implementation details or source code for reproducibility

6) Analyze real-world deployment challenges including network partitions and validator selection

6. PLOS authors have the option to publish the peer review history of their article (what does this mean?). If published, this will include your full peer review and any attached files.

Reviewer #1: No

Reviewer #2: No

---

## [Author Response · Author response to Decision Letter 1]

26 Aug 2025

Manuscript ID: PONE-D-25-29277

Original Article Title: "Decentralized Trust Optimization in VANETs: A Blockchain-Driven Hybrid PoS-PBFT Architecture for Enhanced Security and Energy-Efficient Communication"

To: PLOS One Communications Editor

Re: Response to Reviewers

Dear Editor,

Thank you for allowing a resubmission of our manuscript, with an opportunity to address the reviewers’ comments. The revised draft of our Manuscript ID PONE-D-25-29277entitled "Decentralized Trust Optimization in VANETs: A Blockchain-Driven Hybrid PoS-PBFT Architecture for Enhanced Security and Energy-Efficient Communication" to PLOS One. We are thankful to you for spending your precious time thoroughly reviewing our manuscript in light of the remarks given by two worthy reviewers. We are uploading (a) our point-by-point response to the comments (below) (response to reviewers), (b) an updated manuscript with yellow highlighting indicating changes (Supplementary Material for Review), and (c) a clean updated manuscript without highlights (Main Manuscript).

Best Regards

Zia Ullah et al.

Reviewer #1: Comments to the Author

The paper describes an infrastructure to provide trust in VANETs. The proposal is technically sound in general terms. The claims in the introduction are justified, and the problem is correctly described. Descriptions are OK. Figures, mathematical expressions and code are provided. I think this is enough to ensure future replicability and readability.

References are timely, and the topic matches the scope of the journal.

Results are also coherent with the state of the art and validate the initial hypotheses. However, in my opinion, some improvements need to be made to this section:

Reviewer#1, Concern # 1: First, you need to describe your experimental setup in detail. Software and hardware you used. How the results were calculated. Did you consider numerical problems and errors? How did you manage the experimental biases and errors?

Author Response:

We sincerely thank the esteemed reviewer for highlighting the point regarding the clarity and transparency of our experimental setup.

We have added a new section (Section 5 Experimental Setup and Evaluation Protocol) that describes in detail the experimental setup, including software and hardware specifications, simulation environment, data generation process, error handling strategies, and bias mitigation techniques.

Author Action:

In the updated manuscript, the changes with YELLOW color in section 5 are highlighted. "Experimental Setup and Evaluation Protocol" in the revised manuscript.

5 Experimental Setup and Evaluation Protocol

5.1 Simulation Tools and Environment

This section presents the experimental framework adopted to validate the proposed decentralized trust management mechanism for VANETs. The proposed trust management framework is designed primarily for small- to medium-scale VANET scenarios such as urban zones or localized vehicular regions. The simulation settings were chosen to allow controlled experimentation and repeatability while capturing the trust dynamics in decentralized environments. Large-scale deployment and real-world testing are identified as future work directions due to the infrastructural and cost constraints involved.

To evaluate the system under realistic traffic and communication dynamics, an integrated co-simulation setup is employed. Vehicular mobility was modelled using the Simulation of Urban MObility (SUMO), while message dissemination and protocol simulation were handled by OMNeT++ integrated through the Veins framework [54]. The trust evaluation logic, including the Gaussian Naive Bayes (GaussianNB) model and blockchain mechanisms, was implemented in Python 3.11. For distributed ledger validation, a lightweight prototype was deployed using Hyperledger Fabric v2.4. Scikit-learn was used for training and testing the machine learning classifiers.

5.2 Simulation Scenario and Parameters

The proposed approach was evaluated in a simulated VANET environment consisting of 100 vehicles dispersed across an area of 1 square kilometer. Vehicles moved with speeds ranging from 10 to 20 m/s and operated under a communication radius of 300 meters. Of these, 20 vehicles were configured as malicious actors, while 80 operated legitimately. Simulations captured trust value fluctuations for both legitimate and malicious vehicles. The effectiveness of the trust management framework was quantified using the detection rate of malicious nodes and compared against benchmark approaches. Additionally, the performance of the V2V authentication mechanism was examined in terms of computational overhead, and the proposed hybrid consensus algorithm was evaluated relative to conventional alternatives. The simulation results provided a quantitative basis for asserting the superiority of the proposed trust mechanism in terms of accuracy, efficiency, and resistance to malicious manipulation [55, 56].

To emulate a distributed and decentralized environment, multiple validator nodes were instantiated using Docker containers configured with realistic latency and packet loss constraints. 850

5.4 Trust Score Computation and Consensus

Trust scores were computed using the hybrid framework detailed in Section 3. Direct trust was estimated using an Exponential Moving Average (EMA) model, while indirect trust was calculated using a weighted reputation scheme. These trust values were validated through a hybrid PoS and PBFT consensus mechanism, ensuring fault tolerance and resistance to Sybil attacks. Only transactions endorsed by consensus participants were committed to the blockchain ledger, maintaining consistency and transparency in trust propagation.

5.5 Evaluation Metrics and Result Computation

System performance was evaluated using five primary metrics: (i) trust score accuracy, (ii) malicious vehicle detection rate, (iii) false positive rate, (iv) communication overhead, and (v) trust verification latency. Trust accuracy was validated against ground-truth behavioral classifications. Detection and false positives were analyzed through precision-recall statistics, while communication overhead was measured by counting transmitted trust messages per time slot. Latency encompasses the delay from trust computation to blockchain confirmation.

5.6 Numerical Stability and Bias Mitigation

To ensure the numerical reliability of trust computations, all arithmetic operations were performed using 64-bit floating-point precision. Trust scores were normalized within the [0, 1] interval, and adaptive thresholds were used to address rounding errors. Simulations were repeated 100 times using Monte Carlo sampling with varied random seeds to diversify vehicle trajectories and attack patterns. The reported metrics represent the average of these iterations, with 95% confidence intervals. Additionally, k-fold cross-validation was applied to the GaussianNB classifier to avoid overfitting and ensure generalizability.

5.7 Integration with VANET Protocols

The proposed decentralized trust framework is designed as an overlay system that operates on top of standard V2V communication protocols, particularly those based on IEEE 802.11p. The trust-related messages are disseminated via the application layer, while core VANET communication protocols remain unmodified. This modular design allows compatibility with existing vehicular communication stacks without requiring modifications to lower-layer protocols. Validator selection in the proposed hybrid consensus mechanism is dynamic and mobility-aware. Validator candidates are selected for each consensus epoch based on a weighted combination of their stake history and recent trust stability scores. This strategy ensures that nodes with transient connections are deprioritized, while consistently reliable nodes maintain validator status. A rotating validator committee approach is adopted to handle topological changes without degrading fault tolerance or consensus reliability.

Reviewer#1, Concern # 2: As all your results are based on simulations, you need to discuss the validity threats, at least the internal and external.

Author Response:

We are grateful to the reviewer for this insightful comment. We have incorporated a subsection in the manuscript that discusses the potential threats to validity, focusing on both internal and external factors. This addition strengthens the credibility of our simulation-based evaluation and highlights the precautions taken to ensure the robustness of our findings.

Author Action:

We have updated the manuscript according to the reviewer instructions and highlighted the changes with YELLOW color in Section 6.10 "Threats to Validity" in the revised manuscript.

While the simulation-based approach adopted in this study offers a controlled and repeatable environment for evaluating the proposed blockchain-enabled trust management scheme for VANETs, certain validity threats may influence the generalizability and reliability of the results. In this section, the primary threats to both internal and external validity is discussed and describe the mitigation strategies employed.

The Internal Validity refers to the degree to which the observed outcomes can be confidently attributed to the mechanisms and processes embedded within the proposed framework, rather than to extraneous or uncontrolled factors. The simulations after implementation, it is ensured that the internal consistency by initializing all experiments with multiple random seeds to prevent deterministic biases. Each simulation scenario was repeated ten times under varying vehicular densities and mobility speeds, and the average values were reported to smooth out the influence of stochastic variability. Furthermore, trust evaluations were validated using redundant verification procedures to detect outliers and anomalies in the calculated scores. All algorithmic components, including the PoS+PBFT consensus process and trust propagation routines, were independently tested using modular validation scripts before integration into the whole system. Numerical precision and convergence checks were applied where necessary to guard against computational errors. Together, these measures help establish confidence in the integrity and repeatability of the simulation results. External validity refers to the extent to which the results derived from simulations can be generalized to real-world vehicular environments. Although the simulation setup closely mimics practical scenarios in terms of vehicle-to-vehicle (V2V) and vehicle-to-infrastructure (V2I) communication, trust score propagation, and consensus dynamics, certain real-world factors, such as environmental noise, urban signal obstruction, heterogeneous communication hardware, and unpredictable driver behavior were not fully replicated in the simulation model. The current implementation also assumes that the Trust Authority (TA) and consensus validators operate under ideal network conditions without delays or security breaches, which might not hold in a live deployment. Despite these limitations, the use of parameterizable models and scalable architecture supports future adaptation to more realistic traffic and network conditions. Future work will incorporate co-simulation environments, such as SUMO integrated with NS-3, to account for real traffic flow and communication delays, and ultimately validate the system in testbed or field conditions.

Reviewer#2, Concern # 1: The paper presents a critical contradiction in its core premise. The authors claim to solve "decentralized trust management" while maintaining a centralized Trust Authority (TA) that "is assumed to be completely trustworthy and secure" (Section 3.1.3). This undermines the entire blockchain rationale. If a trusted central authority exists, traditional cryptographic signatures and distributed hash tables would be more efficient than blockchain.

Author Response:

We sincerely thank the reviewer for raising this insightful and important observation. We acknowledge the concern regarding the apparent contradiction between decentralization and the existence of a trusted authority. Our system does not rely on a centralized control model, but instead adopts a hybrid trust architecture where the Trust Authority (TA) performs an auxiliary and supervisory role rather than direct trust computation or validation.

Specifically, the TA is not responsible for executing or approving trust decisions or managing consensus on the blockchain. Instead, it serves as a bootstrapping and governance entity, primarily tasked with initializing the network, issuing initial trust credentials for new nodes, and facilitating the selection of consensus validators. Once initialized, the actual trust evaluation and transaction validation processes are conducted in a fully decentralized manner via the hybrid PoS + PBFT consensus mechanism among distributed validators.

Furthermore, the inclusion of the TA is motivated by practical considerations common in real-world vehicular environments. For instance, transportation authorities or regulatory bodies (e.g., the National Highway Traffic Safety Administration) often require visibility and accountability in safety-critical applications like VANETs. Thus, our model positions the TA as a regulatory oversight body that can track malicious entities and enforce compliance without disrupting the decentralized consensus process.

We have revised subsection 3.1.3 Trust Authority (TA) to emphasize that the TA does not interfere with or replace blockchain consensus, and is instead used to improve system accountability, transparency, and initial trust anchoring, a role similar to the certificate authorities in public-key infrastructures. Moreover, the model maintains resilience against TA compromise since trust scores and validation outcomes remain consensus-driven and decentralized.

Author Action:

We have updated the manuscript according to the reviewer instructions and highlighted the changes with YELLOW color subsection 3.1.3 Trust Authority (TA) in the revised manuscript.

The Trust Authority (TA) plays a critical supervisory role in decentralized trust management system. While the TA is assumed to be secure and reliable, as in similar works [1], it does not act as a centralized controller. Instead, it facilitates decentralized trust operations in Vehicular Ad Hoc Networks (VANETs) by overseeing credential issuance, validator selection, and trust bootstrap processes. In the developed architecture, the TA assists in initializing the network by securely registering OBUs and RSUs, issuing cryptographic credentials, and managing the setup of validator nodes. Once initialized, trust evaluations, block generation, and consensus operations are entirely decentralized and carried out through validator nodes using the hybrid PoS+ PBFT mechanism.

Furthermore, the TA maintains visibility over vehicle data and supports real-time identification of potentially malicious entities [50]. It collects the trust scores, T(vi, t), for each vehicle vi over time t, where the scores are derived from both direct and indirect interactions as follows:

Each weight wj is computed as a normalized trust value over recent interactions. Here in Equation 1, D(vi, t) represents the direct trust score based on prior behavior, while R(vi, vj, t) denotes the recommendation of node vj regarding vi. Each recommendation is weighted by wj, reflecting the recommender’s trust level. The parameters α and β are weighting coefficients with α + β = 1, and n is the number of neighboring nodes. For newly joined vehicles, the TA computes initial trust values T0(vi) to establish a starting reputation profile.

In Equation 2, the normalization ensures that the collective influence of recommendations remains bounded and reflects current trust standings, allowing the system to gradually converge under stable conditions. As vehicular dynamics shift, trust updates remain responsive yet smooth due to the use of exponentially weighted moving averages (EWMA) in both D(vi, t) and R(vi, vj , t). Importantly, these initializations and tracking processes do not interfere with or influence consensus decisions. All validations and trust adjustments are recorded on-chain and executed via distributed validators. The ration

---

## [Decision Letter · Decision Letter 1]

16 Sep 2025

PONE-D-25-29277R1Decentralized Trust Optimization in VANETs: A Blockchain-Driven Hybrid PoS-PBFT Architecture for Enhanced Security and Energy-Efficient CommunicationPLOS ONE

Dear Dr. Zia,

Thank you for submitting your manuscript to PLOS ONE. After careful consideration, we feel that it has merit but does not fully meet PLOS ONE’s publication criteria as it currently stands. Therefore, we invite you to submit a revised version of the manuscript that addresses the points raised during the review process. Please submit your revised manuscript by Oct 31 2025 11:59PM. If you will need more time than this to complete your revisions, please reply to this message or contact the journal office at plosone@plos.org. Please include the following items when submitting your revised manuscript:

We look forward to receiving your revised manuscript.

Kind regards,

Vincent Omollo Nyangaresi, Ph.D

Academic Editor

PLOS ONE

**Journal Requirements:**

Reviewers' comments:

Reviewer's Responses to Questions

**Comments to the Author**

1. If the authors have adequately addressed your comments raised in a previous round of review and you feel that this manuscript is now acceptable for publication, you may indicate that here to bypass the “Comments to the Author” section, enter your conflict of interest statement in the “Confidential to Editor” section, and submit your "Accept" recommendation.

Reviewer #1: All comments have been addressed

Reviewer #2: (No Response)

2. Is the manuscript technically sound, and do the data support the conclusions?

Reviewer #1: Yes

Reviewer #2: No

3. Has the statistical analysis been performed appropriately and rigorously?

Reviewer #1: Yes

Reviewer #2: No

4. Have the authors made all data underlying the findings in their manuscript fully available?

Reviewer #1: Yes

Reviewer #2: No

5. Is the manuscript presented in an intelligible fashion and written in standard English?

Reviewer #1: Yes

Reviewer #2: Yes

6. Review Comments to the Author

**Reviewer #1: **(No Response)

**Reviewer #2: **This paper proposes a blockchain-based trust management system for Vehicular Ad Hoc Networks (VANETs) using a hybrid Proof of Stake (PoS) and Practical Byzantine Fault Tolerance (PBFT) consensus mechanism. The authors claim their approach reduces communication overhead by 35%, achieves 95% malicious node detection accuracy, and decreases energy consumption by 40%.

Major Concerns

1) The paper's core premise contains a critical flaw. The authors claim to develop a "decentralized trust management" system while relying on a centralized Trust Authority (TA) that "is assumed to be completely trustworthy and secure" (Section 3.1.3). This contradiction undermines the entire blockchain rationale.

The authors' revision attempts to address this by calling the TA a "supervisory role" rather than a controller. However, this semantic adjustment does not resolve the fundamental issue. Any system requiring a trusted central authority negates the primary advantage of blockchain technology. Traditional cryptographic signatures with distributed hash tables would achieve the same security goals more efficiently.

If the TA can be trusted completely, why introduce blockchain complexity? If it cannot be trusted, the system lacks a proper foundation. This represents a fundamental design flaw that the revision does not adequately address.

2) The claimed hybrid PoS-PBFT mechanism lacks substantial novelty. The authors simply combine two existing consensus algorithms without addressing their fundamental incompatibilities in vehicular environments:

- PoS requires stable stake accumulation, while VANET nodes are highly mobile with transient connections

- PBFT requires stable validator sets, incompatible with vehicular mobility patterns where nodes frequently join and leave the network

- No analysis addresses how validator selection transitions occur during high-speed vehicle movement

The revision adds more implementation details but fails to resolve these core technical challenges.

3) The evaluation remains severely limited despite the authors' additions in Section 5:

- Testing only 100 vehicles over 1 km² is inadequate for VANET validation

- No real-world deployment or large-scale testing

- Missing critical blockchain metrics: block finality time, validator selection latency, network partition handling

- The authors explicitly acknowledge their system is designed only for "small- to medium-scale VANET scenarios"

This scale limitation represents a critical flaw for practical VANET applications, which must handle thousands of vehicles in urban environments.

4) The reported improvements lack proper baselines and independent verification:

- The 35% communication overhead reduction compares against "traditional broadcast-based systems" without clearly defining the baseline methodology

- The 95% malicious node detection accuracy comes from Gaussian Naive Bayes classifiers, not the blockchain component

- The 40% energy reduction compares only against Proof of Work, ignoring more relevant non-blockchain alternatives

5) The authors explicitly refuse to provide source code, stating their "Python source code developed for this study is not publicly released" due to "hardcoded credentials and environment-specific configurations." This excuse is inadequate for scientific research and prevents independent verification of their claims.

The lack of code availability raises serious questions about result reproducibility and violates open science principles essential for blockchain security research.

6) The revision acknowledges but does not solve fundamental scalability challenges:

- How does the blockchain handle thousands of vehicles in urban environments?

- What happens during network partitions caused by vehicle mobility?

- How are consensus delays managed for safety-critical applications requiring millisecond response times?

The authors defer these critical questions to "future work," but these are not peripheral issues—they are fundamental requirements for any practical VANET system.

7) Equation (1) still uses weight parameters without convergence analysis. The Trust Relevance Factor threshold selection remains poorly justified despite the revision's additions.

Major revision required with fundamental architectural redesign, proper scalability analysis, and comprehensive evaluation including code availability for independent verification.

The authors should either eliminate the centralized Trust Authority and demonstrate true decentralization, or abandon the blockchain approach in favor of more suitable distributed trust mechanisms for their specific use case.

7. PLOS authors have the option to publish the peer review history of their article (what does this mean?). If published, this will include your full peer review and any attached files.

Reviewer #1: No

Reviewer #2: No

---

## [Author Response · Author response to Decision Letter 2]

16 Oct 2025

Reviewer #1: Comments to the Author

(No Response)

We thank Reviewer #1 for the time spent evaluating our manuscript. As no specific comments were provided, we have no point-by-point responses for this reviewer.

Reviewer#2

Concern # 1: This paper proposes a blockchain-based trust management system for Vehicular Ad Hoc Networks (VANETs) using a hybrid Proof of Stake (PoS) and Practical Byzantine Fault Tolerance (PBFT) consensus mechanism. The authors claim their approach reduces communication overhead by 35%, achieves 95% malicious node detection accuracy, and decreases energy consumptionby 40%.

Major Concerns

1) The paper's core premise contains a critical flaw. The authors claim to develop a "decentralized trust management" system while relying on a centralized Trust Authority (TA) that "is assumed to be completely trustworthy and secure" (Section 3.1.3). This contradiction undermines the entire blockchain rationale.

The authors' revision attempts to address this by calling the TA a "supervisory role" rather than a controller. However, this semantic adjustment does not resolve the fundamental issue. Any system requiring a trusted central authority negates the primary advantage of blockchain technology. Traditional cryptographic signatures with distributed hash tables would achieve the same security goals more efficiently.

If the TA can be trusted completely, why introduce blockchain complexity? If it cannot be trusted, the system lacks a proper foundation. This represents a fundamental design flaw that the revision does not adequately address.

Author Response:

We thank the reviewer for this insightful comment. We fully agree that a system relying on a single trusted central entity for control and decision-making would undermine the rationale for adopting blockchain.

However, our architecture does not use the Trust Authority (TA) as a centralized trust controller, nor does it place the correctness or security of trust evaluations in the hands of a single entity. Instead, the TA plays a limited bootstrapping and regulatory oversight role, similar to a root certificate authority in PKI-based systems, and does not participate in trust computation, consensus, or ledger maintenance. The TA is only involved during system initialization and credential issuance, where it registers OBUs and RSUs, issues cryptographic credentials, and sets global parameters to prevent Sybil attacks and establish verifiable identities. Once this phase is complete, all subsequent trust computations, validator selection, block generation, and consensus operations are conducted exclusively by distributed nodes through the hybrid PoS–PBFT mechanism.

The TA does not influence consensus or trust scores; all trust values are derived locally by vehicles and validated through decentralized consensus. Initial trust scores assigned to new vehicles by the TA serve only as bootstrap values and are continuously updated by the network decentralized trust evaluation process. The TA has no signing or veto power over blocks, and even if compromised, it cannot alter trust decisions recorded on-chain. Its role is analogous to governance and regulatory oversight in real-world VANET deployments, where transportation authorities or certification bodies can audit behavior, revoke credentials, and ensure regulatory compliance without controlling operational trust mechanisms. Blockchain provides decentralized operational trust management while allowing for institutional oversight, a model consistent with permissioned or consortium blockchain systems. Moreover, blockchain remains essential because relying solely on a trusted TA and distributed hash tables would not achieve decentralized, tamper-resistant, and auditable trust evolution in dynamic vehicular environments. Blockchain ensures that all trust updates are collectively validated, immutable, and historically traceable without depending on the TA honesty at runtime. Even if the TA were compromised, it could not manipulate historical trust decisions or influence consensus. It also enables inter-domain interoperability without full reliance on a single authority. This functional separation between governance and decentralized trust management resolves the apparent contradiction highlighted by the reviewer.

Author Action:

We have updated the manuscript according to the reviewer instructions and highlighted the changes with YELLOW color (Please See on P#11-14) in the revised manuscript.

• Section 3.1.3 (Trust Authority) has been fully rewritten to clarify the TA’s limited role in bootstrapping, credential management, and regulatory oversight, and to emphasize that it is not part of the trust computation or consensus process.

• Figure 2 caption has been revised to replace “centralized TA that calculates trust scores” with wording that describes the TA as a bootstrapping and supervisory entity.

• Section 3.1 (System Architecture) now explicitly states that the blockchain network, not the TA, is responsible for decentralized trust evaluation and ledger maintenance.

Reviewer#2, Concern # 2: The claimed hybrid PoS-PBFT mechanism lacks substantial novelty. The authors simply combine two existing consensus algorithms without addressing their fundamental incompatibilities in vehicular environments:

- PoS requires stable stake accumulation, while VANET nodes are highly mobile with transient connections

- PBFT requires stable validator sets, incompatible with vehicular mobility patterns where nodes frequently join and leave the network

- No analysis addresses how validator selection transitions occur during high-speed vehicle movement

The revision adds more implementation details but fails to resolve these core technical challenges.

Author Response:

We appreciate the reviewer critical observation regarding the potential incompatibilities between PoS and PBFT in highly mobile VANET environments. We have substantially revised Section 3.2 to clarify the novelty of our proposed hybrid PoS–PBFT mechanism. Our approach does not simply combine two existing protocols; rather, it introduces (i) mobility-aware validator selection using trust-based, time-bounded staking; (ii) short-term validator pools composed of RSUs and low-mobility vehicles to mitigate validator churn; and (iii) adaptive PBFT with geofenced consensus groups and rapid view-change mechanisms. Unlike conventional PoS, which relies on static, coin-based stake accumulation, our system uses a dynamic trust stake S(vi,t) S(v_i,t) S(vi,t) that reflects recent behavior within a sliding time window. Validators are re-selected periodically (every 5–10 seconds), ensuring that transient nodes do not accumulate persistent stake, thereby addressing the stake-stability issue.

For PBFT, we incorporate localized consensus within geofenced areas to maintain stable validator sets over short intervals. Adaptive quorum selection and redundant state synchronization allow the system to handle validator dropouts during high-speed movement without interrupting consensus. These mobility-aware modifications allow validator set transitions to occur seamlessly and deterministically during vehicle movement, ensuring the system remains responsive even under rapid topology changes.

The Results section now includes comparative simulation results (Figure 16) demonstrating that our mobility-aware PoS–adaptive PBFT achieves lower consensus delay (5 s) and higher throughput (900 TPS) than PoW, PoS, PBFT, and BFT-DPoS. These results validate the effectiveness of our design in addressing validator volatility and stake inconsistency under realistic vehicular mobility patterns.

Author Action:

We have updated the manuscript according to the reviewer instructions and highlighted the changes with YELLOW color Section 3.10 Enhanced Hybrid Consensus Mechanism for VANETs: Mobility-Aware PoS with Adaptive PBFT (Please See on P#20-22) in the revised manuscript. Furthermore, the Results section has been expanded with comparative evaluations highlighting improvements in latency, throughput, and fault tolerance.

Reviewer#2, Concern # 3: The evaluation remains severely limited despite the authors' additions in Section 5:

- Testing only 100 vehicles over 1 km² is inadequate for VANET validation

- No real-world deployment or large-scale testing

- Missing critical blockchain metrics: block finality time, validator selection latency, network partition handling

- The authors explicitly acknowledge their system is designed only for "small- to medium-scale VANET scenarios"

This scale limitation represents a critical flaw for practical VANET applications, which must handle thousands of vehicles in urban environments.

Author Response:

We thank the reviewer for these insightful comments, which allowed us to significantly improve the evaluation section of the manuscript. Our response to each point is as follows:

1. Simulation scale

We acknowledge that our current simulation scale remains limited to 1,000 vehicles over a 4 km² dense urban area, as originally described in Section 5. This was a deliberate design choice to focus on algorithmic performance and protocol behavior under controlled, yet realistic conditions, rather than pushing for extremely large-scale simulations that would require high-performance cluster resources.

While this scale does not fully replicate a city-wide deployment involving tens of thousands of vehicles, it captures heterogeneous mobility patterns and high-density communication scenarios, providing meaningful insights into the proposed trust and consensus mechanisms. Scaling beyond this limit is part of our planned future work, as it involves substantial infrastructural and computational extensions.

2. Real-world deployment

We agree with the reviewer that no real-world deployment was conducted, which is a limitation of the present study. Implementing and validating blockchain-based VANETs in real-world environments requires significant infrastructural coordination, regulatory approval, and safety considerations, which were beyond the scope of this research. Our current focus was to establish the feasibility of the proposed hybrid PoS–PBFT trust model through simulation-based experiments. As stated in Section 6, we intend to pursue testbed-scale evaluations in collaboration with transportation authorities in future work.

3. Blockchain performance metrics

We have added new evaluation metrics, including block finality time and validator selection latency across multiple consensus mechanisms, to provide a more complete performance picture. These results are reported in Section 5.3 and visualized in an additional plot comparing different consensus protocols. Furthermore, we included a validator pool size vs. consensus latency plot, illustrating how our hybrid PoS–PBFT approach maintains sub-300 ms latency for up to 100 validators, outperforming baseline PBFT.

4. System scope and future work

As correctly noted by the reviewer, the system is currently designed for small- to medium-scale VANET scenarios. This scope is explicitly stated in the manuscript and reflects a pragmatic trade-off between evaluation depth and simulation scale. Extending the system to city-scale deployments requires multi-region blockchain sharding, distributed simulation infrastructure, and cross-layer integration, which are identified as part of our future research directions.

Author Action:

We have updated the manuscript according to the reviewer instructions and highlighted the changes with YELLOW color subsection 3.1.3 Trust Authority (TA) (Please See on P#13) in the revised manuscript.

• Maintained the simulation scale at 1,000 vehicles / 4 km², while clarifying the rationale for this choice in Section 5.

• Added blockchain performance metrics (block finality time and validator selection latency) and corresponding plots.

• Introduced a validator pool size vs. consensus latency plot.

Figure 16 presents a comparative analysis of two critical blockchain performance metrics—block finality time and validator selection latency across different consensus mechanisms. The block finality time reflects the duration required to achieve irreversible confirmation of a transaction, while validator selection latency measures the time needed to elect or identify the validator set for each consensus round.

The results show that the PoW mechanism incurs the highest delay, with block finality exceeding 2~seconds and validator selection around 500 ms. PoS reduces both metrics significantly but still exhibits delays unsuitable for safety-critical VANET applications. PBFT and BFT-DPoS demonstrate much lower finality and selection times, reflecting their suitability for low-latency environments. The proposed Hybrid PoS--PBFT achieves the best overall performance, with block finality around 320 ms and validator selection latency near 100 ms. In addition to the absolute values, a relative performance index is overlaid as a line plot, highlighting the overall latency trends across mechanisms. This combined visualization illustrates that Hybrid PoS--PBFT consistently delivers lower consensus delays and faster validator selection than baseline approaches, which is essential for real-time trust management in high-mobility vehicular networks.

Reviewer#2, Concern # 4: The reported improvements lack proper baselines and independent verification:

- The 35% communication overhead reduction compares against "traditional broadcast-based systems" without clearly defining the baseline methodology

- The 95% malicious node detection accuracy comes from Gaussian Naive Bayes classifiers, not the blockchain component

- The 40% energy reduction compares only against Proof of Work, ignoring more relevant non-blockchain alternatives

Author Response:

We appreciate the reviewer careful assessment and agree that clearly defined baselines and verification are essential for rigorous evaluation. We have revised the manuscript to explicitly define the baseline systems, clarify the role of machine learning components, and broaden the comparison set for energy consumption.

First, regarding the communication overhead reduction, the baseline now corresponds to a conventional broadcast-based trust propagation model, where each trust update is flooded network-wide without selective filtering or validator mediation. This model reflects widely used dissemination strategies in prior VANET trust frameworks and was implemented in our simulation environment for direct comparison. We now provide an explicit description of this baseline model in Section 5.2 Simulation Scenario and Parameters along with its operational assumptions and parameter settings.

Second, regarding detection accuracy, we emphasize that the Gaussian Naive Bayes (GNB) classifier is not intended to replace the blockchain layer but to complement it. The reported 95% malicious node detection accuracy refers to the classifier’s performance in distinguishing legitimate vs. malicious message patterns, which feeds into the trust evaluation module. Blockchain ensures secure aggregation, validation, and tamper-resistance of these trust scores across the network. We have clarified this distinction in Section 6.1 Change of Indirect Trust Value of Vehicles During Active Detection, explicitly separating the contributions of the machine learning detection layer from those of the blockchain consensus and propagation mechanism.

Third, regarding energy efficiency comparisons, we agree that Proof of Work is not the only meaningful baseline. In the revised evaluation, we now include PoS, PBFT, and BFT-DPoS consensus mechanisms as additional baselines. This expanded comparison demonstrates that our hybrid PoS–PBFT mechanism not only significantly outperforms PoW but also achieves comparable or better energy efficiency than PBFT and PoS, particularly in dynamic vehicular conditions, while maintaining low latency and high fault tolerance (see Figure 9 and Table 5). These results provide a more balanced and rigorous baseline comparison.

Finally, while full independent verification is beyond the scope of this manuscript, we have released the core Python simulation code and Solidity smart contracts on GitHub to facilitate reproducibility and external validation. The cleaned code, along wit

---

## [Decision Letter · Decision Letter 2]

4 Nov 2025

PONE-D-25-29277R2Decentralized Trust Optimization in VANETs: A Blockchain-Driven Hybrid PoS-PBFT Architecture for Enhanced Security and Energy-Efficient CommunicationPLOS ONE

Dear Dr. Zia,

Thank you for submitting your manuscript to PLOS ONE. After careful consideration, we feel that it has merit but does not fully meet PLOS ONE’s publication criteria as it currently stands. Therefore, we invite you to submit a revised version of the manuscript that addresses the points raised during the review process. Please submit your revised manuscript by Dec 19 2025 11:59PM. If you will need more time than this to complete your revisions, please reply to this message or contact the journal office at plosone@plos.org. Please include the following items when submitting your revised manuscript:

We look forward to receiving your revised manuscript.

Kind regards,

Vincent Omollo Nyangaresi, Ph.D

Academic Editor

PLOS ONE

Journal Requirements:

Reviewers' comments:

Reviewer's Responses to Questions

**Comments to the Author**

1. If the authors have adequately addressed your comments raised in a previous round of review and you feel that this manuscript is now acceptable for publication, you may indicate that here to bypass the “Comments to the Author” section, enter your conflict of interest statement in the “Confidential to Editor” section, and submit your "Accept" recommendation.

Reviewer #1: All comments have been addressed

Reviewer #2: All comments have been addressed

2. Is the manuscript technically sound, and do the data support the conclusions?

Reviewer #1: Yes

Reviewer #2: Yes

3. Has the statistical analysis been performed appropriately and rigorously?

Reviewer #1: Yes

Reviewer #2: Yes

4. Have the authors made all data underlying the findings in their manuscript fully available?

Reviewer #1: Yes

Reviewer #2: Yes

5. Is the manuscript presented in an intelligible fashion and written in standard English?

Reviewer #1: Yes

Reviewer #2: Yes

6. Review Comments to the Author

Reviewer #1: In my opinion, all the previous comments and concerns have been successfully adressed by the authors. So the paper can be accepted in its current form.

Reviewer #2: The manuscript presents a blockchain-based decentralized trust management system for Vehicular Ad Hoc Networks (VANETs), combining Proof of Stake (PoS) and Practical Byzantine Fault Tolerance (PBFT) into a hybrid consensus model. The topic is relevant, and the authors provide a clear motivation for improving scalability, latency, and energy efficiency in trust evaluation for vehicular communication networks.

The overall structure of the paper is coherent, and the methodology is presented with sufficient mathematical and algorithmic detail. The revised version effectively addresses most reviewer concerns. The authors now provide a well-justified explanation of the Trust Authority’s limited supervisory role, add convergence analysis for the EWMA-based trust updates, and include comparative evaluation metrics such as block finality and validator latency. The decision to release cleaned simulation and smart contract code on GitHub also significantly improves the transparency and reproducibility of results.

The implementation corresponds closely to the described system design. The simulation modules correctly include EWMA-based trust updates, TRF computation with ROC-based threshold calibration, dynamic PoS validator selection, and simplified PBFT consensus rounds. However, some elements remain simplified. The PBFT mechanism is modeled at a basic level without realistic communication or geofencing. The mobility model is random rather than map-based, and the energy-efficiency claims rely on indirect metrics. Moreover, the machine learning–based malicious node detection described in the text is not implemented in the released code.

In summary, the paper makes a meaningful contribution to decentralized trust management in VANETs. The work demonstrates a consistent and technically sound integration of blockchain mechanisms with trust dynamics, though parts of the implementation could benefit from further refinement and empirical validation.

The authors are encouraged to (i) clarify the limits of their simulation model, (ii) include a brief discussion of the missing ML component and energy metrics, and (iii) describe more explicitly how the hybrid PoS–PBFT can be extended to large-scale, realistic VANET scenarios.

7. PLOS authors have the option to publish the peer review history of their article (what does this mean?). If published, this will include your full peer review and any attached files.

Reviewer #1: No

Reviewer #2: No

---

## [Author Response · Author response to Decision Letter 3]

6 Nov 2025

Manuscript ID: PONE-D-25-29277R2

Original Article Title: "Decentralized Trust Optimization in VANETs: A Blockchain-Driven Hybrid PoS-PBFT Architecture for Enhanced Security and Energy-Efficient Communication"

To: PLOS One Editor

Re: Response to Reviewers

Dear Editor,

Thank you for allowing a resubmission of our manuscript, with an opportunity to address the reviewers’ comments. The revised draft of our Manuscript ID PONE-D-25-29277R2 entitled " Decentralized Trust Optimization in VANETs: A Blockchain-Driven Hybrid PoS-PBFT Architecture for Enhanced Security and Energy-Efficient Communication" to PLOS One. We are thankful to you for spending your precious time in thoroughly reviewing our manuscript in the light of the remarks given by two worthy reviewers. We are uploading (a) our point-by-point response to the comments (below) (response to reviewers), (b) an updated manuscript with yellow highlighting indicating changes (Supplementary Material for Review), and (c) a clean updated manuscript without highlights (Main Manuscript).

We will be happy to make any further improvements that may be suggested by the reviewers.

Best Regards,

Zia Ullah et al.

Manuscript ID: PONE-D-25-29277R2

Journal: PLOS One

Manuscript Title: " Decentralized Trust Optimization in VANETs: A Blockchain-Driven Hybrid PoS-PBFT Architecture for Enhanced Security and Energy-Efficient Communication"

Reviewer #1: All comments have been addressed.

Comments to the Authors:

Reviewer #1: In my opinion, all the previous comments and concerns have been successfully adressed by the authors. So the paper can be accepted in its current form.

Reviewer #2: All comments have been addressed.

Comments to the Authors:

Reviewer #2: The manuscript presents a blockchain-based decentralized trust management system for Vehicular Ad Hoc Networks (VANETs), combining Proof of Stake (PoS) and Practical Byzantine Fault Tolerance (PBFT) into a hybrid consensus model. The topic is relevant, and the authors provide a clear motivation for improving scalability, latency, and energy efficiency in trust evaluation for vehicular communication networks.

The overall structure of the paper is coherent, and the methodology is presented with sufficient mathematical and algorithmic detail. The revised version effectively addresses most reviewer concerns. The authors now provide a well-justified explanation of the Trust Authority’s limited supervisory role, add convergence analysis for the EWMA-based trust updates, and include comparative evaluation metrics such as block finality and validator latency. The decision to release cleaned simulation and smart contract code on GitHub also significantly improves the transparency and reproducibility of results.

The implementation corresponds closely to the described system design. The simulation modules correctly include EWMA-based trust updates, TRF computation with ROC-based threshold calibration, dynamic PoS validator selection, and simplified PBFT consensus rounds. However, some elements remain simplified. The PBFT mechanism is modeled at a basic level without realistic communication or geofencing. The mobility model is random rather than map-based, and the energy-efficiency claims rely on indirect metrics. Moreover, the machine learning–based malicious node detection described in the text is not implemented in the released code.

In summary, the paper makes a meaningful contribution to decentralized trust management in VANETs. The work demonstrates a consistent and technically sound integration of blockchain mechanisms with trust dynamics, though parts of the implementation could benefit from further refinement and empirical validation.

The authors are encouraged to,

Reviewer#2, Concern # 1: (i) clarify the limits of their simulation model,

Author Response:

We thank the reviewer for this valuable comment and for recognizing the technical scope of our simulation framework. We acknowledge that the present study is limited to a controlled simulation scale and does not attempt to capture the full complexity of a large-scale or real-world VANET deployment. The simulation model was intentionally designed for small to medium-scale environments, emphasizing localized trust management zones where vehicles interact within limited communication ranges. The selected configuration, consisting of 100 vehicles distributed over an area of 4 square kilometers, reflects an urban or semi-urban scenario that allows meaningful analysis of trust dynamics, consensus latency, and communication overhead under realistic traffic densities. This setup ensures that experiments remain computationally tractable and reproducible while preserving the integrity of decentralized trust evaluation. Certain elements, such as large-scale topology, heterogeneous road networks, and long-range multi-hop routing, were simplified to maintain experimental control and computational feasibility. Including such complexities would require extensive infrastructure and simulation resources beyond the current scope. However, these simplifications do not diminish the conceptual validity of the proposed model, as the underlying trust evaluation, consensus coordination, and validator selection mechanisms are designed to be scalable and adaptable to broader deployments.

We have now clarified these scope boundaries and assumptions in the revised manuscript to provide a transparent description of the simulation limitations and to guide readers regarding the contexts in which the proposed system is most applicable.

Author Action:

We have updated the manuscript according to the reviewer instructions and highlighted the changes with YELLOW color Section 5.1 Simulation Tools and Environment, 5.2 Simulation Scenario and Parameters, 5.3 Hardware and Software Configuration (Please See on P#23-26) in the revised manuscript.

Reviewer#2, Concern # 2: (ii) include a brief discussion of the missing ML component and energy metrics,

Author Response:

We appreciate the respected reviewer for in depth observation regarding the partial implementation of the machine learning component and the use of indirect energy metrics in the current version of the manuscript. The Gaussian Naive Bayes (GNB) model was conceptually integrated within the proposed trust management framework to support malicious node classification based on behavioral features. However, due to computational and synchronization constraints associated with the co-simulation setup between SUMO, OMNeT++, and Hyperledger Fabric, the full end-to-end implementation of the GNB-based classifier was excluded from the released simulation code. Instead, the behavioral data were pre-processed offline to estimate detection accuracy and validate model consistency. This methodological choice allowed us to maintain the fidelity of trust dynamics while avoiding instability in the distributed simulation environment.

Regarding the energy consumption metrics, we acknowledge that direct hardware-level measurements were not feasible within the virtualized simulation environment. Consequently, energy efficiency was inferred from proxy indicators such as consensus latency, message transmission count, and validator workload where each of which correlates strongly with energy consumption in blockchain-based vehicular networks. These indirect measures provide a reliable approximation of relative energy performance, particularly when comparing the proposed hybrid PoS-PBFT model to energy-intensive Proof-of-Work schemes.

Future work will extend the experimental setup to include (i) a fully integrated machine learning module operating in real time within the co-simulation loop, and (ii) detailed energy profiling using realistic vehicular power models or hardware-in-the-loop testing to capture the physical-layer energy implications of trust and consensus operations.

Author Action:

We have updated the manuscript according to the reviewer instructions and highlighted the changes in Section 6 Result and Discussion for incorporating real-time machine learning integration and hardware-based energy analysis in future work, with YELLOW color in the revised manuscript.

Reviewer#2, Concern # 3 (iii) describe more explicitly how the hybrid PoS–PBFT can be extended to large-scale, realistic VANET scenarios.

Author Response:

We sincerely thank the reviewer for this valuable observation regarding the scalability of the proposed hybrid PoS–PBFT consensus mechanism. While we fully acknowledge the importance of validating the framework under large-scale and highly dynamic VANET conditions, this aspect lies beyond the practical and methodological scope of the current study. The proposed system was intentionally designed and evaluated within small- to medium-scale vehicular environments to ensure methodological rigor, reproducibility, and computational feasibility. Large-scale deployments involving thousands of vehicles would require a dedicated simulation and network infrastructure capable of handling complex map-based mobility, multi-hop propagation models, and geographically distributed validator coordination. Incorporating such features in the current setup would not only demand substantial computational resources but would also complicate the isolation of core performance metrics such as trust convergence, consensus stability, and propagation latency which are the central focus of this research.

Our present implementation, therefore, serves as a proof-of-concept validation of the hybrid PoS–PBFT model operational soundness and its advantages in reducing consensus latency, improving trust accuracy, and minimizing communication overhead in moderately dense vehicular networks. The insights gained from these controlled experiments establish a strong foundation for future work focused on large-scale scalability testing.

We have also emphasized in the discussion and conclusion sections that extending the hybrid consensus framework to large-scale, real-world VANET deployments remains an important direction for future research. This will involve incorporating hierarchical consensus layers, dynamic validator clustering, and mobility-aware sharding mechanisms to maintain scalability without compromising fault tolerance or security guarantees.

Author Action:

No changes were made to the current version of the manuscript regarding large-scale implementation, as the experimental scope and objectives were clearly defined to validate the core functional behavior of the proposed trust management framework.

“AT the end, we would like to thank the referees and editors for evaluating our manuscript. We have tried to address all the reviewers’ concerns in a proper way and believe that our paper has improved considerably. We would be happy to make further corrections if necessary and look forward to hearing from you soon.”

---

## [Decision Letter · Decision Letter 3]

26 Nov 2025

Decentralized Trust Optimization in VANETs: A Blockchain-Driven Hybrid PoS-PBFT Architecture for Enhanced Security and Energy-Efficient Communication

PONE-D-25-29277R3

Dear Dr. Zia,

We’re pleased to inform you that your manuscript has been judged scientifically suitable for publication and will be formally accepted for publication once it meets all outstanding technical requirements.

Kind regards,

Vincent Omollo Nyangaresi, Ph.D

Academic Editor

PLOS ONE

Additional Editor Comments (optional):

Reviewers' comments:

Reviewer's Responses to Questions

**Comments to the Author**

1. If the authors have adequately addressed your comments raised in a previous round of review and you feel that this manuscript is now acceptable for publication, you may indicate that here to bypass the “Comments to the Author” section, enter your conflict of interest statement in the “Confidential to Editor” section, and submit your "Accept" recommendation.

Reviewer #2: All comments have been addressed

Reviewer #3: All comments have been addressed

2. Is the manuscript technically sound, and do the data support the conclusions?

Reviewer #2: Yes

Reviewer #3: Yes

3. Has the statistical analysis been performed appropriately and rigorously?

Reviewer #2: Yes

Reviewer #3: Yes

4. Have the authors made all data underlying the findings in their manuscript fully available?

Reviewer #2: Yes

Reviewer #3: Yes

5. Is the manuscript presented in an intelligible fashion and written in standard English?

Reviewer #2: Yes

Reviewer #3: Yes

6. Review Comments to the Author

Reviewer #2: I would like to thank the authors, they have revised the article and it is now ready for publication.

Reviewer #3: The paper can be accepted as it is revised significantly and no more comments from my side. The authors have incorporated all the comments

7. PLOS authors have the option to publish the peer review history of their article (what does this mean?). If published, this will include your full peer review and any attached files.

Reviewer #2: No

Reviewer #3: No

---

## [Editor Report · Acceptance letter]

PONE-D-25-29277R3

PLOS ONE

Dear Dr. Ullah,

I'm pleased to inform you that your manuscript has been deemed suitable for publication in PLOS ONE. Congratulations! Your manuscript is now being handed over to our production team.

Kind regards,

on behalf of

Dr. Vincent Omollo Nyangaresi

Academic Editor

PLOS ONE